# A child with perinatal HIV infection and long-term sustained virological control following antiretroviral treatment cessation

Avy Violari[1], Mark F. Cotton[2], Louise Kuhn[3], Diana B. Schramm[4,5], Maria Paximadis[4,5], Shayne Loubser[4,5], Sharon Shalekoff[4,5], Bianca Da Costa Dias[4,5], Kennedy Otwombe[1], Afaaf Liberty[1], James McIntyre[6,7], Abdel Babiker[8], Diana Gibb[8] & Caroline T. Tiemessen[4,5]

Understanding HIV remission in rare individuals who initiated antiretroviral therapy (ART) soon after infection and then discontinued, may inform HIV cure interventions. Here we describe features of virus and host of a perinatally HIV-1 infected child with long-term sustained virological control. The child received early limited ART in the Children with HIV Early antiRetroviral therapy (CHER) trial. At age 9.5 years, diagnostic tests for HIV are negative and the child has characteristics similar to uninfected children that include a high CD4:CD8 ratio, low T cell activation and low CCR5 expression. Virus persistence (HIV-1 DNA and plasma RNA) is confirmed with sensitive methods, but replication-competent virus is not detected. The child has weak HIV-specific antibody and T cell responses. Furthermore, we determine his *HLA* and *KIR* genotypes. This case aids in understanding post-treatment control and may help design of future intervention strategies.

[1] Perinatal HIV Research Unit, Faculty of Health Sciences, University of the Witwatersrand, PO Box 114, Diepkloof, Soweto 1864, South Africa. [2] Family Clinical Research Unit, Department of Paediatrics and Child Health, Stellenbosch University, PO Box 241, Cape Town 8000, South Africa. [3] Gertrude H. Sergievsky Center, College of Physicians and Surgeons; and Department of Epidemiology, Mailman School of Public Health, Columbia University Medical Center, 630 W 168th Street, New York, NY 10032, USA. [4] Centre for HIV and STIs, National Institute for Communicable Diseases (NICD), of the National Health Laboratory Service (NHLS), Johannesburg 2131, South Africa. [5] Faculty of Health Sciences, University of the Witwatersrand, Parktown, Johannesburg 2193, South Africa. [6] Anova Health Institute, 12 Sherborne Road, Parktown, Johannesburg 2193, South Africa. [7] School of Public Health and Family Medicine, Faculty of Health Sciences, University of Cape Town, Anzio Road, Observatory 7925, South Africa. [8] Medical Research Council, Clinical Trials Unit, University College London, 90 High Holborn, London WC1V 6LJ, UK. Correspondence and requests for materials should be addressed to A.V. (email: violari@mweb.co.za) or to C.T.T. (email: carolinet@nicd.ac.za)

Rapid formation of persistent viral reservoirs follows acute HIV-1 infection. This early establishment of latently HIV-1-infected CD4+ T cells harbouring replication-competent virus remains the major obstacle to HIV cure or remission[1–3]. As antiretroviral therapy (ART), even when given within days of infection, usually fails to clear these reservoirs[4–6], it is unlikely that ART alone can lead to HIV remission. It is, however, hypothesized that ART given very soon after infection may enable a more effective immune response and, together with other strategies, lead to sustained control of viral replication.

Current approaches to HIV cure or remission have focused on either reversing latency (e.g. "shock and kill"), enhancing immune responses or preventing immune activation (e.g. vaccines and other immunotherapies)[7]. Central to the question of HIV remission is the interaction between viral reservoir, immune activation, host genetics and immune response.

Several adult cases of post-treatment control have been described[8–16]. These individuals are unlike elite controllers (<1%) who control HIV-1 to undetectable levels in the absence of ART[17,18], probably through distinct immunological mechanisms[8].

In children, data are extremely limited. In 2013, the report of the "Mississippi baby" suggested that very early ART, here within 30 h of birth, could lead to prolonged (27 months) virological control off-treatment[19,20], raising hope for a feasible HIV-1 remission strategy. Unfortunately, this girl "relapsed" after almost 2 years without ART due to return of high levels of viral replication, and required ART. Subsequently, a French girl was reported who started ART at 3 months of age, stopped treatment between 5 and 7 years of age and controlled virus to undetectable levels for over 12 years[21].

Reports of post-treatment controllers who initiated ART and then discontinued by design or unintentionally may help our understanding of key host determinants of HIV replication control, and inform interventions for HIV remission and cure.

Here we report a detailed virological and immunological analysis of a child at 9.5 years of age, originally enroled in the Children with HIV Early antiRetroviral therapy (CHER) trial[22,23] who was randomized to the immediate, time-limited 40 weeks of ART study arm. The CHER trial was initiated at a time when the best strategy on when to initiate and how to maintain treatment in infants was unclear. This child, one of 227 early treated children (0.4%), is the only one maintaining long-term sustained virological control post-ART cessation. At 9.5 years, virus persists at low levels (plasma RNA 6.6 copies per mL), cell-associated DNA is 5 copies per million peripheral blood mononuclear cells and replication-competent virus is not detected. Immunologically, he is not unlike healthy children of similar age, evidenced by high CD4:CD8 ratio, low T cell activation and low CCR5 expression. He has weak HIV-specific antibody and CD4+ T cell responses indicating memory of prior/current virus encounter, and together with possession of some host genotypes, these provide clues for future studies to inform what constitutes long-term post-treatment control.

## Results

**Clinical case.** The child, born in 2007, had a positive HIV-1 DNA PCR at age 32 days. At 39 days, HIV-1 RNA was >750,000 copies per mL (upper limit of quantitation of the assay) confirming infection; at 60 days, plasma HIV RNA had declined to 151,000 copies per mL. He commenced zidovudine, lamivudine and lopinavir-ritonavir one day later (Fig. 1, Supplementary Table 1). He was born at term, of normal birth weight (3700g), did not receive nevirapine prophylaxis, and was not breastfed. CD4+ T cell count and per cent at 61 days, prior to ART start, were 2249

cells per μL and 41.6%. These values fell within the respective baseline interquartile ranges (IQRs) for all early treated children who stopped ART in the CHER trial (Supplementary Table 2)—$n = 227$ children; median CD4+ T cell count was 2255 (IQR: 1759–2972); median CD4% 36.4 (IQR: 31.4–42.5).

Viral load (VL) declined to <50 RNA copies per mL after 24 weeks of ART. At 50 weeks of age, when treatment was stopped per trial protocol, VL was <20 copies per mL. Thereafter, VL remained below detection over 8.75 years without ART (Fig. 1a). CD4+ T cell counts remained normal-for-age (Fig. 1b) and CD4% remained above 30% throughout (Fig. 1c). At 9.5 years plasma drug concentrations for the most commonly used antiretroviral agents in South Africa were undetectable. The mother's CD4+ T cell count was 108 cells per μL when he was 7 months of age, and then 129 cells per μL 20 months later—these are the only maternal data available.

**Virus persistence.** At 9.5 years, the Roche VL result was reported as target not detected (TND; Fig. 2a). Virus pelleted from 10 mL of plasma, yielded 66 copies and a VL of 6.6 RNA copies per mL (Fig. 2a). Using a highly sensitive RNA nested qPCR on 3 mL of plasma, 2 of 22 replicates were positive (Fig. 2a). Using a semi-nested real-time PCR (sn-qPCR) assay and an input of 1 μg of genomic DNA (gDNA) per well, total cell-associated HIV-1 DNA was estimated at 5 copies per $10^6$ peripheral blood mononuclear cells (PBMCs; six of nine amplifications positive) (Fig. 2b). In a stored sample from ART interruption at 50 weeks of age, this was almost identical (5 copies per $10^6$ PBMCs: 1 of 3 amplifications positive). DNA sequencing of *gag* from the 9.5 year sample confirmed infection with subtype C virus (Fig. 2c). Using two virus outgrowth assays (primary CD8-depleted PBMCs and MOLT4/CCR5 cells), no replication-competent virus was detected at 9.5 years (Fig. 2d). However, a weak HIV RNA signal (1 of 24 replicates was positive) was detected in the day 28 supernatant of cultured primary CD4+ cells from the 50-week sample using qualitative real-time PCR nested assay (n-qPCR). The child's CD4+ T cells could be infected in vitro with the HIV-1 BaL strain (Fig. 2d).

**HIV-specific antibodies.** At 9.5 years of age, HIV-specific antibodies were undetectable by enzyme-linked immunosorbent assay (ELISA) (optical density (OD) 0.056; cut-off 0.263), and the western blot was indeterminate with weak reactivity to Gag proteins (p24++, p40++ and p55/51+) (Fig. 3a). By multiplex bead arrays the child showed a substantial IgA2 response to gp41, and weak but detectable responses to gp120 and Vpu (IgG1), Gag (IgG2), Tat (IgG2, IgG3, IgA1) and Vpu9 peptide (IgM) (Fig. 3b).

**Cellular immune responses.** The child possessed strong T cell responses to staphylococcal enterotoxin B (SEB; 9.03% CD4, 6.35% CD8) and a very strong anti-CD16-induced natural killer (NK) cell response (27.79% interleukin-2/interferon-γ (IL-2/IFN-γ)+CD3−CD56+ cells) (Fig. 3c). A weak (0.116%) CD4+ T cell response to Gag was detected, without detectable CD8+ T cell responses to any peptide pool (Fig. 3d).

**Host genotyping.** All *HLA* loci were heterozygous: A*30:02:01/66:01; B*08:01:01/44:03:01; C*04:01:01/07:01:01; DPB1*01:01:01/18:01; DQB1*05:01:01/06:09:01; DRB1*12:01:01/13:02:01 (Fig. 4a). The child's *KIRAA1* genotype, the most common genotype globally, included both full-length (f) and truncated (v) *KIR2DS4* alleles (Fig. 4b). These are among features that we have associated with greater risk of transmission/acquisition of HIV-1 infection in mother-to-child transmission studies[24,25]. *HLA B*44:03:01* has a threonine (T) at position 80 as part of the Bw4

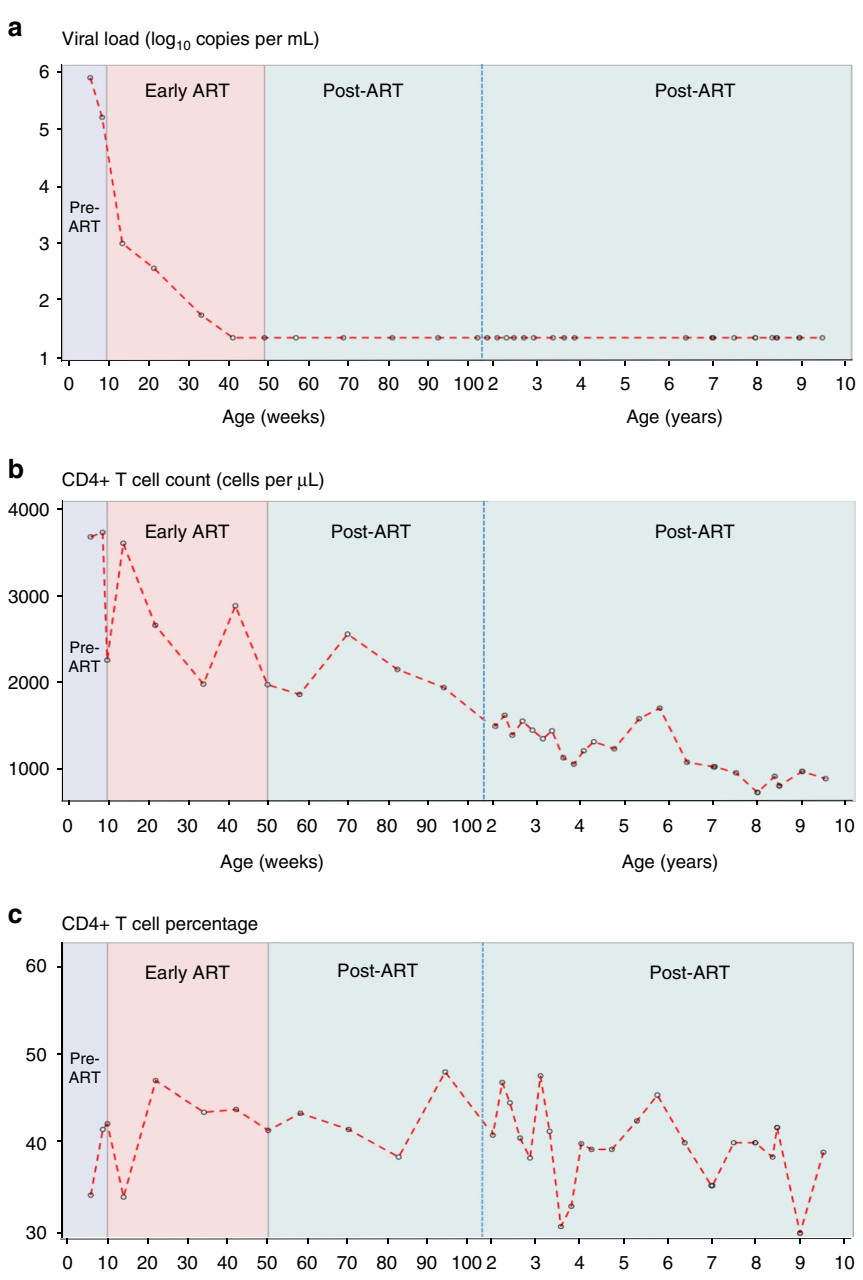

**Fig. 1** Longitudinal viral loads and CD4+ T cell counts. **a–c** Viral load (log$_{10}$ RNA copies per mL) (**a**), absolute CD4+ T cell count (cells per μL) (**b**) and CD4 + T cell percentage (**c**) are shown over time up to age 9.5 years (open circles joined by red dotted lines). Pre-ART, early ART and post-ART periods are indicated (shaded in lavender, pink, pale green, respectively). The X-axis shows age in weeks and then in years, separated by a blue vertical dotted line. Note: the detection limit of the VL assays used was 400 RNA copies per mL at 10.71 and 116.71 weeks; 50 RNA copies per mL at 33.71 weeks and 20 RNA copies per mL at all other time points. ART antiretroviral therapy

epitope (NLRTALR) of the α1 helix 2 of the molecule—and therefore will bind to KIR3DL1 on NK cells. HLA- Cw*07:01:01 (allotype C1) and Cw*04:01:01 (C2) have a lysine (K) and asparagine (N) at position 80, respectively—allowing for inter- action with both KIR2DL3 (C1) and 2DL1 (C2) on NK cells. Collectively, these genotyping data suggest the potential for diverse interactions that may include engagement of CD4 and CD8 T cells and NK cells in antiviral responses.

**Immunophenotyping.** The CD4:CD8 T cell ratio was 1.9, higher than all uninfected control children (Fig. 5a). Measurement of T cell subsets representing various stages of differentiation

highlighted that 9.5–10-year-old HIV-uninfected children do not display adult-like proportions of the different subsets, particularly for naive, central memory and effector memory CD4+ T cell subsets and central memory CD8+ T cells (Fig. 5b). The child had high proportions of naive, central memory and effector memory CD8+ T cells compared to uninfected children.

CCR5 density on CD4+ and CD8+ T cells was similar for children and adults, with the child having lower CCR5 expression than 14 of the 15 uninfected children/adults (Fig. 5c). Proportions of the child's CCR5-expressing CD4+ and CD8+ T cells were comparable to those of uninfected children. Of note, CCR5- expressing CD8+ T cells were significantly higher in adults than

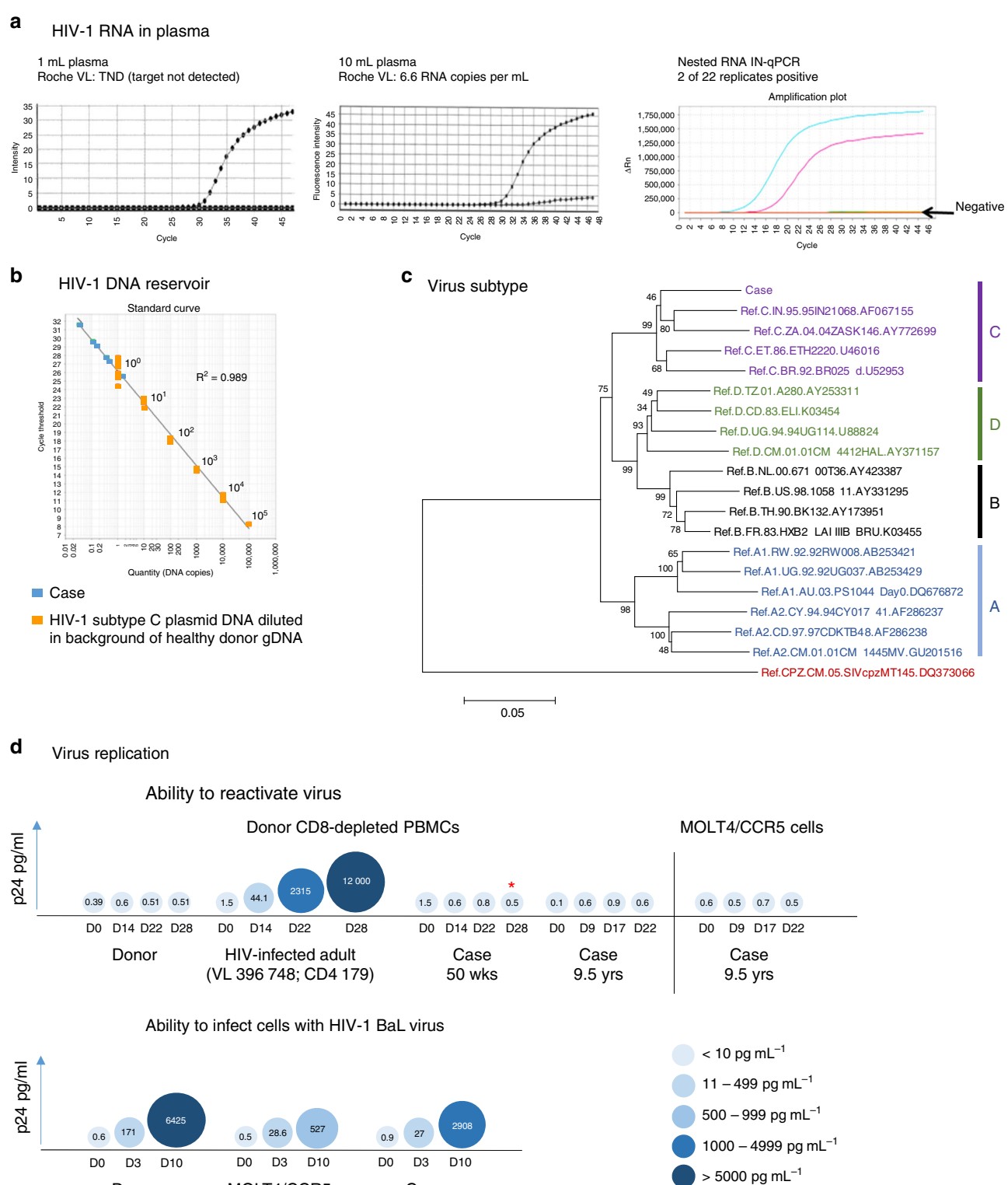

children ($p = 0.0013$; Mann–Whitney $U$ test). Levels of immune activation, measured by HLA-DR, were similar to those of uninfected children and adults. Expression of TIGIT in the child's CD4+ and CD8+ T cells was similar to uninfected children, and significantly higher in adults ($p = 0.027$ and $p = 0.0027$, respectively; Mann–Whitney $U$ tests). PD-1 expression on the child's CD4+ T cells was substantially above the median values for children or adults. However, two adult outliers had similarly high levels. Similar to TIGIT, PD-1 expression on CD8+ T cells was significantly higher in adults than in children ($p = 0.0007$; Mann–Whitney $U$ test), with the child having high levels similar to the adult median. Flow cytometry data are provided in Supplementary Table 3.

## Discussion

We report virological and immunological characteristics in a South African child of 9.5 years of age in long-term HIV

**Fig. 2** Detection of HIV RNA, HIV DNA and replication-competent virus. **a** Viral load results at 9.5 years of age when testing the standard 1 mL of plasma (target not detected (TND); left) vs. 10 mL (middle) by the standard Roche assay. A qualitative ultrasensitive RNA nested integrase PCR (IN-qPCR) assay conducted on 3 mL plasma; DNAse treated to ensure no contaminating cell-associated HIV-1 DNA (right). **b** Quantitation of the total HIV-1 DNA reservoir using a semi-nested quantitative reverse transcription PCR (RT-qPCR) assay at 9.5 years. The standard curve (orange squares) shows plasmid copy number controls (1–100,000 copies) on the x-axis and corresponding cycle threshold values on the y-axis. The case replicates are shown as blue squares. Curves lower than the $10^0$ (1 copy) plasmid control are counted as 1 copy. **c** A neighbour-joining phylogenetic tree constructed using the partial gag-PR sequence (1414 bp HXB2 nt 903–2334; Gag aa39–501, PR aa 1–28). Reference subtypes A–D (in blue, black, purple and green, respectively) are included and the tree is rooted on SIV chimpanzee sequence (Los Alamos HIV sequence database; https://www.hiv.lanl.gov). Accession numbers (e.g. AF067155) of reference sequences are indicated in the figure. Numbers at the nodes indicate percentage bootstrap scores (n = 1000). The child's (Case) gag consensus sequence (see Supplementary Fig. 1) is indicated and clusters with subtype C sequences (purple). **d** The ability to reactivate virus from the child's CD4+ T cells was measured using two co-culture methods: donor CD8-depleted peripheral blood mononuclear cells (PBMCs) and MOLT4/CCR5 cells (top of panel). Included is the healthy HIV-1-uninfected donor (negative control) and an HIV-1-infected patient with high viral load, CD4+ T cell count <200 cells per μL (positive control). The ability to infect CD4+ cells from the case with HIV-1 BaL (bottom of panel). Donor and MOLT4/CCR5 cells were included as positive controls. The case was tested at the indicated times (age). The different sizes and shades of blue colour of the circles represent the p24 concentration in culture supernatants; the actual pg mL$^{-1}$ values appear within the circles (the colour key shows ranges of levels of p24 according to shade of blue, with p24 levels increasing with increasing intensity of colour). *Indicates the time point at which a very weak signal was obtained by ultrasensitive nested RNA IN-qPCR assay in the 50-week sample

remission. He initiated ART at 8.7 weeks of age and discontinued ART after 40 weeks in accordance to his randomization in the CHER trial[22]. This is the only child to achieve this outcome among 227 who stopped ART (at 40 or 96 weeks) in the trial (0.4%). The French case of long-term remission was one of 15 children in the French paediatric cohort who stopped ART (6.7%)[21].

Studies have shown that earlier ART initiation results in a smaller HIV reservoir size[26–28]. Treatment was very early (within 30 h of birth) for the Mississippi baby who achieved 27 months of virological control off-treatment before experiencing virological rebound[19,20]. This delay in rebound may have been attributed to a small size of latent replication-competent reservoir. Both the French case with >12 years of remission[21] and this South African case started ART later, at 3 and 2 months, respectively. HIV transmission was likely intrapartum in the French child and in utero in the Mississippi baby. The timing is unknown for the South African child. Timing of transmission may have been a key factor affecting the different outcomes of remission. Of note, subtype of virus (B, H and C, respectively), treatment duration (18 months, 6 years, 10 months, respectively) and ethnicity were different in these children. Both earlier cases were girls, while this is the first report of a boy with HIV remission. Unlike the South African child, using similar methods and number of CD4+ T cells, replication-competent virus was readily found in the French case[21]. In contrast, no replication-competent virus was found in the Mississippi baby when testing 22 million CD4+ T cells[19], highlighting that this measure poorly predicts long-term likelihood of remission. Furthermore, recent reports of very early treatment in adult cases (within 2 weeks of infection) described multiple sensitive tests for HIV-1 persistence, which supported the absence of detectable virus during ART—these cases all rebounded shortly after ART was stopped (n = 8 patients[5]; median 26 days, range 13–48 days; n = 1 patient[6], 225 days). Collectively, these findings suggest some limited viral replication may in fact be required for durable long-term remission.

Early in infection the child's VLs were high, indicative of a highly replicating virus. The VL declined from >750,000 to 150,000 RNA copies per mL prior to beginning treatment, suggesting an immune response attempting to control virus replication. After ART initiation, viral decline was biphasic, with the expected initial sharp decline followed by a more gradual decline. After ART cessation, VLs remained below the detection threshold for 8.75 years.

More sensitive methods for VL measurement confirmed the presence of low amounts of virus produced in vivo (6.6 RNA copies per mL). Inability to detect replication-competent virus in vitro may be because of assay sensitivity (2 million CD4+ cells tested), or defective virus that cannot accumulate to detectable levels.

Early initiation of ART is associated with non-reactive HIV antibody results in many HIV-1-infected children and adults[29–31]. On standard assays the child is seronegative; however, results from bead arrays revealed footprints of historical adaptive responses that have either waned or are being maintained through ongoing antigenic priming. The substantial IgA2 (mucosal) response to gp41 could be primed by microbial antigens sharing homology with gp41[32–34]. However, a recent study reported stronger gp41-specific IgA responses in elite controllers, which could not be well explained by responses to microbial antigens[35]. HIV-1 Env- and Vpu-specific NK cell antibody-dependent cellular cytotoxicity (ADCC) responses, including against C-terminal peptide Vpu19 (our Vpu9 peptide), are associated with elite HIV-1 control[36]. Using a whole blood assay, we have shown similar responses associated with reduced maternal–infant HIV-1 transmission and lower VLs in HIV-1-infected mothers[37,38]. The child lacked a detectable NK cell response to any HIV-1 peptide pool, likely due to very low levels of antibodies that target Env and Vpu. The Vpu9-IgM antibody response suggests a potential for interaction with complement. Anti-Tat IgG responses have also been associated with natural HIV-1 control and improved immune function in ART-treated patients receiving Tat vaccine[39–42]. Interestingly, an African study highlighted that persistent anti-Tat IgM in addition to IgG might be protective against disease progression[41].

An IgG2 antibody response to Gag has been associated with CD4+ T cell response and long-term nonprogression[43]. The child had both responses. The weak Gag-specific CD4+ T cell response without a detectable CD8+ T cell response is intriguing. Such responses are reported in HIV-exposed uninfected individuals and some HIV-uninfected individuals, only when using sensitive cultured ELISpot assays[44]. However, the opposite exists in untreated HIV-1-infected infants where we readily demonstrated Gag-specific CD8+, but not CD4+ HIV-specific T cell responses in the first few months of life—using the same whole blood intracellular cytokine assay as for the present study[45]. Furthermore, there is an absence of sustained HIV-specific T cell responses in early ART-treated HIV-1-infected children[46]. The CD4+ Gag response demonstrated in the child seems remarkable given early treatment and long duration of viral suppression. The CD4+ T cell response may be maintained by small amounts of virus (replication-competent or replication-defective) or Gag protein produced in vivo. Lack of CD8+ T cell response suggests that CD8+ T cells might not be currently involved in controlling levels of viraemia, supporting the possibility that HIV may be

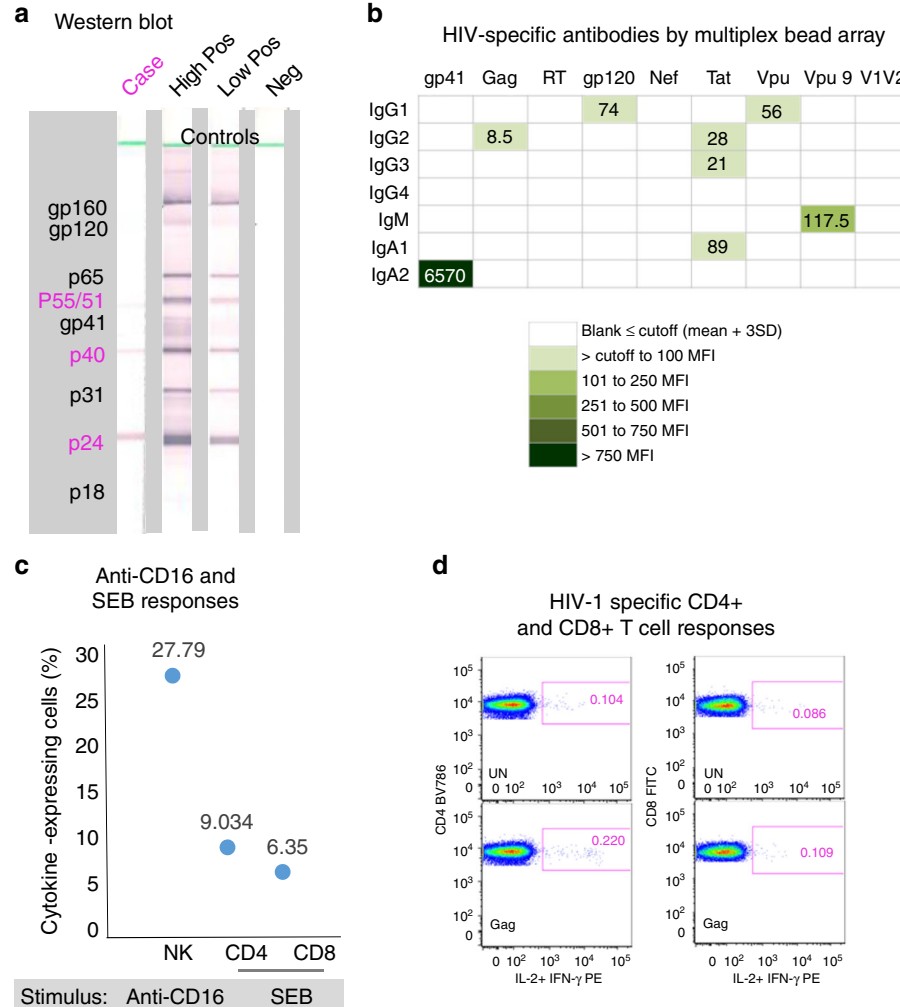

**Fig. 3** HIV-specific responses and immune response capability of the case at 9.5 years of age. **a** Detection of HIV-specific antibodies at 9.5 years of age by western blot. The case antibody profile is compared with controls that are a high positive, low positive and HIV-negative. HIV proteins corresponding to bands in the blots are shown in the left grey-shaded block; the case profile was positive for the core proteins indicated in pink. **b** Quantitation of HIV-specific antibodies by multiplex bead array for all isotypes and subclasses (indicated on the left side—IgG1, IgG2, IgG3, IgG4, IgM, IgA1, IgA2) against gp41, Gag, RT, gp120, Nef, Tat, Vpu, peptide Vpu9 and V1V2 scaffold antigens (indicated at the top). Results are expressed as mean fluorescence intensities (MFI)), the colour key shows ranges of MFI according to colour intensity (the darker the more HIV-specific antibody detected). A result is considered positive if above the cut-off (mean ± 3 SD) determined from eight adult uninfected controls. Vpu9 amino acid sequence: STMVDMGHLRLLDVNDL. **c** Proportions of natural killer (NK) cells that respond to anti-CD16 antibody, and CD4+ and CD8+ T cells that respond to staphylococcal enterotoxin B (SEB) in a whole blood intracellular cytokine (ICC) assay that measures induction of interferon-γ (IFN-γ) and interleukin-2 (IL-2). HIV-uninfected adult reference values for comparison (n = 21; median % and range)—natural killer (NK) anti-CD16%: 37.92 (12–67.6), CD4 SEB%: 6.04 (0.25–11.91), CD8 SEB%: 5.82 (0.18–18.94). **d** A weak positive CD4+ T cell response to Gag (0.116%) in the absence of a detectable CD8+ T cell response to Gag (<0.1%; 0.023%). UN: addition of costimulatory antibodies anti-CD28 and anti-CD49d, no stimulation with peptides

defective for further infection of permissive cells. In keeping with this, we hypothesize the HIV-1 reservoir in the child may be maintained through homeostatic, antigen-stimulated or integration site-dependent proliferation of CD4+ T cells harbouring transcription/translation-competent but not necessarily replication-competent HIV-1. These cellular proliferative mechanisms present a challenge for HIV-1 eradication strategies[47]. In contrast, the French case showed weak but broader CD4+ and CD8+ T cell responses to Gag, Pol and Nef, and with readily achievable viral proliferation in culture[21].

These collective findings raise questions of whether cross-reacting antigens might, in addition to small amounts of virus production in vivo, contribute to maintaining some of these memory responses—and, if such responses contributed to remission in this child. Importantly, lack of detection does not preclude having CD8+ T cell responses[48], or other protective responses such as ADCC/phagocytosis in the presence of maternal/child antibody, that may have been active early in life and possibly essential to the outcome of HIV-1 remission. The rapid decline in VL over a month prior to ART initiation may be of considerable importance in understanding the reasons behind virological control in the child. ART may have protected an early pre-ART antiviral response that may otherwise have been compromised by continued viral replication.

The child had no HLA class I alleles shown in adults to associate with HIV-1 control[49], except for heterozygosity at all HLA loci that is considered advantageous[50]. In contrast, the French case was homozygous at three loci, considered disadvantageous[21]. The child possesses HLA class II alleles already associated with the robust mucosal CD4+ T cell responses in elite

**a**

| | | HLA class I | | | | | HLA class II | | |
|---|---|---|---|---|---|---|---|---|---|
| | A* | A allotype | B* | B allotype | Cw* | C allotype | DPB1* | DQB1* | DRB1* |
| Allele 1 | 30:02:01 | Non-Bw4 | 08:01:01 | Bw6 | 04:01:01 | C2 | 01:01:01 | 05:01:01 | 12:01:01 |
| Allele 2 | 66:01 | Non-Bw4 | 44:03:01 | Bw4-80TA | 07:01:01 | C1 | 18:01 | 06:09:01 | 13:02:01 |

**b**

| | | | | | | KIR genotype | | | | | | | | | | | |
|---|---|---|---|---|---|---|---|---|---|---|---|---|---|---|---|---|---|
| 2DL1 | 2DL2 | 2DL3 | 2DL4 | 2DL5 | 2DS1 | 2DS2 | 2DS3 | 2DS4 | 2DS5 | 2DP1 | 3DL1 | 3DL2 | 3DL3 | 3DS1 | 3DP1 | KIR # | Genotype |
| ■ | | ■ | ■ | | | | | f/v | | ■ | ■ | ■ | ■ | | ■ | 9 | AA1 |

**Fig. 4** *HLA* class I and II alleles and *KIR* genotype. **a** *HLA* class I A, B and C with corresponding allotype designations (non-Bw4, Bw6, Bw4-80TA, C1 and C2), and *HLA* class II DPB1, DQB1 and DRB1 alleles are indicated as allele 1 and 2. *HLA B*44:03:01* has a threonine (T) at position 80 as part of the Bw4 epitope (NLRTALR) of the α1 helix 2 of the molecule—and therefore will bind to KIR3DL1 on NK cells. HLA- Cw*07:01:01 (allotype C1) and Cw*04:01:01 (C2) have a lysine (K) and asparagine (N) at position 80, respectively—allowing for interaction with both KIR2DL3 (C1) and 2DL1 (C2) on NK cells. **b** The presence (coloured box) or absence of 16 *KIR* genes (14 functional and 2 pseudogenes *2DP1* and *3DP1*) is shown. The child has an *AA1* genotype, which has 9 *KIR* genes, has 2 copies of *KIR3DL1* and 2 copies of activating gene *KIR2DS4* with one full-length (*f*) and one truncated (*v*) version (heterozygous *f/v*). 2D: 2-domain; 3D: 3-domain; L: long tail, inhibitory; S: short tail, activating

controllers (DQB1*06 and DRB1*13:02:01)[51]. DQB1*06 has also been associated with IgA responses to V1V2 and increased risk of acquisition in the RV144 HIV vaccine trial[52]. Possession of these class II alleles may explain why the CD4+ Gag response remains detectable at 9.5 years, despite the low-level antigenic exposure. *KIR* genotyping revealed some features already associated with greater risk of vertical transmission in mother-to-child HIV transmission studies[24,25].

The child had a healthy CD4:CD8 T cell ratio and levels of immune activation similar to healthy uninfected children of similar age. Robust T cell and NK cell responses to stimuli suggested a good immune response capability. CCR5 density on T cells was amongst the lowest when compared to HIV-uninfected children or adults—a feature that may be advantageous. Although the child's PD-1-expressing CD4+ T cell proportions were higher than children and adults, PD-1-expressing CD8+ T cells were comparable to adults but much higher than children. High PD-1 expression is unlikely from immune exhaustion given no evidence of high immune activation in the child unlike in chronic untreated and virologically suppressed ART-treated HIV-1 infection[53,54]. In this regard TIGIT, another marker of immune exhaustion, is not different from that in uninfected children. Overall, these features show an immune system that closely resembles that of an uninfected child of similar age, making this child an ideal example of post-treatment HIV-1 control.

The ability of this child to both control virus to levels below detection on standard assays and to control immune activation presents the best of both worlds; these features have been described separately for elite controllers[18,55], non-pathogenic nonhuman primate Simian immunodeficiency virus infection and long-term non-progressing children[56,57]. Events that may have led to sufficient silencing of virus replication will be explored by whole-genome sequencing of virus/provirus. Further investigation will expand our understanding of how the immune system controls HIV-1 replication with accompanying low immune activation, and inform future strategies for ART interruption and other interventions for HIV remission.

## Methods

**The case and study participants.** In the CHER trial (ClinicalTrials.gov Identifier: NCT00102960; 2005–2011), HIV-1-infected treatment-naive, asymptomatic infants aged ≤12 weeks with a CD4% ≥25% were randomized to early limited ART for 40 (ART-40W; *n* = 143) or 96 (ART-96W; *n* = 143) weeks or to deferred ART (*n* = 125)[23]. In the ART-40W group, 122 met criteria to stop ART, together with 105 in the ART-96W group. Re-initiation of ART was based on a CD4% decrease to <20% and on clinical criteria[23]. By the trial end, median follow-up was 4.9 years (IQR: 3.7–5.3). The child in this report was randomized to the ART-40W arm.

Follow-up after the trial continued as part of PEPFAR-supported routine health services and later in an observational study[58,59]. CD4+ T cell counts were determined using the FACSCount System (BD Biosciences, San Jose, CA, USA).

Plasma HIV-1 RNA levels were quantitated retrospectively on stored samples by Roche Ampliprep/COBAS® Taqman HIV-1 Test, v2.0 (Roche Molecular Systems, Inc., Branchburg, NJ, USA) with a lower detection limit of 20 RNA copies per mL (except for three time points where detection limits of assays used were 50 or 400 RNA copies per mL).

To gain further insight into specific characteristics of this case, detailed virological, immunological and genetic studies were undertaken at 9.5 years of age. Some viral studies were conducted on stored samples from 50 weeks of age at ART interruption. The mother is deceased and no stored maternal samples are available for study.

For comparisons of immune cell phenotypes, we included samples from five healthy age-matched HIV-uninfected children (3 females, 2 males; median 9 years, range 9.5–10 years) and 10 HIV-uninfected adults (5 females, 5 males; median age 44 years, range 35–55 years) from the same population.

**Ethics statement.** The CHER trial was approved by the Ethics Committees of the University of the Witwatersrand and Stellenbosch University. Thereafter, the Human Research Ethics Committee of the University of the Witwatersrand provided approval for all subsequent observational studies and investigations of the Case. All participants provided written informed consent. For all minors, a parent or guardian gave written informed consent and the child gave written assent. We have complied with all ethical regulations.

**Assays to detect HIV-1 DNA and RNA.** An sn-qPCR assay targeting reverse transcriptase (RT) was developed for subtype C proviral DNA quantitation based on methods of Pasternak et al[60] and Kiselinova et al[61], and described in Kuhn et al[28]. The sn-RT-qPCR is a two-step PCR in which the first round of amplification was carried out using conventional PCR and allowed to proceed for 15 cycles only and the total product of the first-round PCR was subsequently used in the second round of PCR, a real-time hydrolysis probe-based PCR using a fluorescently labelled TaqMan probe, one primer identical to the forward primer used in the first-round PCR and a second reverse primer designed to bind "deeper" within the amplicon. The second-round PCR was carried out for the standard cycle numbers used in real-time PCR (40 cycles). The numbers of HIV-1 proviral DNA copies were determined by using the standard curve method. To construct the standard curve, known copies of linearized HIV-1 p8.MJ4 plasmid DNA were serially diluted and amplified in a background of HIV-1-negative human gDNA (the same amount used in the experimental wells) and subjected to the same cycling conditions. Sequences of primers and probe are as follows: first-round PCR, sn-RT-forward primer-1 5'-CAT TTC TTT GGA TGG GGT ATG A-3' and sn-RT-reverse primer-1 5'-CCT GTT CTC TGC CAA TTC TAA TTC TGC-3'; second-round qPCR, forward primer identical to sn-RT-forward primer-1 above and sn-RT-reverse primer-2; 5'-TTG CCC AGT TTA ATT TTC CCA CTA-3'; sn-RT-probe; 5'–6 FAM-AGC TGG ACT GTC AAT GA-MGB-3'. gDNA was extracted using the QIAamp DNA Blood kit (Qiagen, Hilden, Germany). Conventional PCR was performed using the Roche Expand High-Fidelity (HiFi) PCR System (Roche Diagnostics GmbH, Roche Applied Science, Mannheim, Germany) and the qPCR was carried out using LightCycler 480 Probes Master Mix (Roche Diagnostics). Cycling conditions were standard for both Expand HiFi and TaqMan hydrolysis probe qPCR. At 9.5 years of age, a total of 9 μg of gDNA extracted from isolated PBMCs (cell equivalent is $1.36 \times 10^6$ cells) was tested at an input of 1 μg per well. The standard curves were run in triplicate at each plasmid dilution, except for the 1 copy standard where five replicates were done. At 50 weeks of age, 3 μg of PBMC DNA (cell equivalent is $4.54 \times 10^5$ cells) was tested.

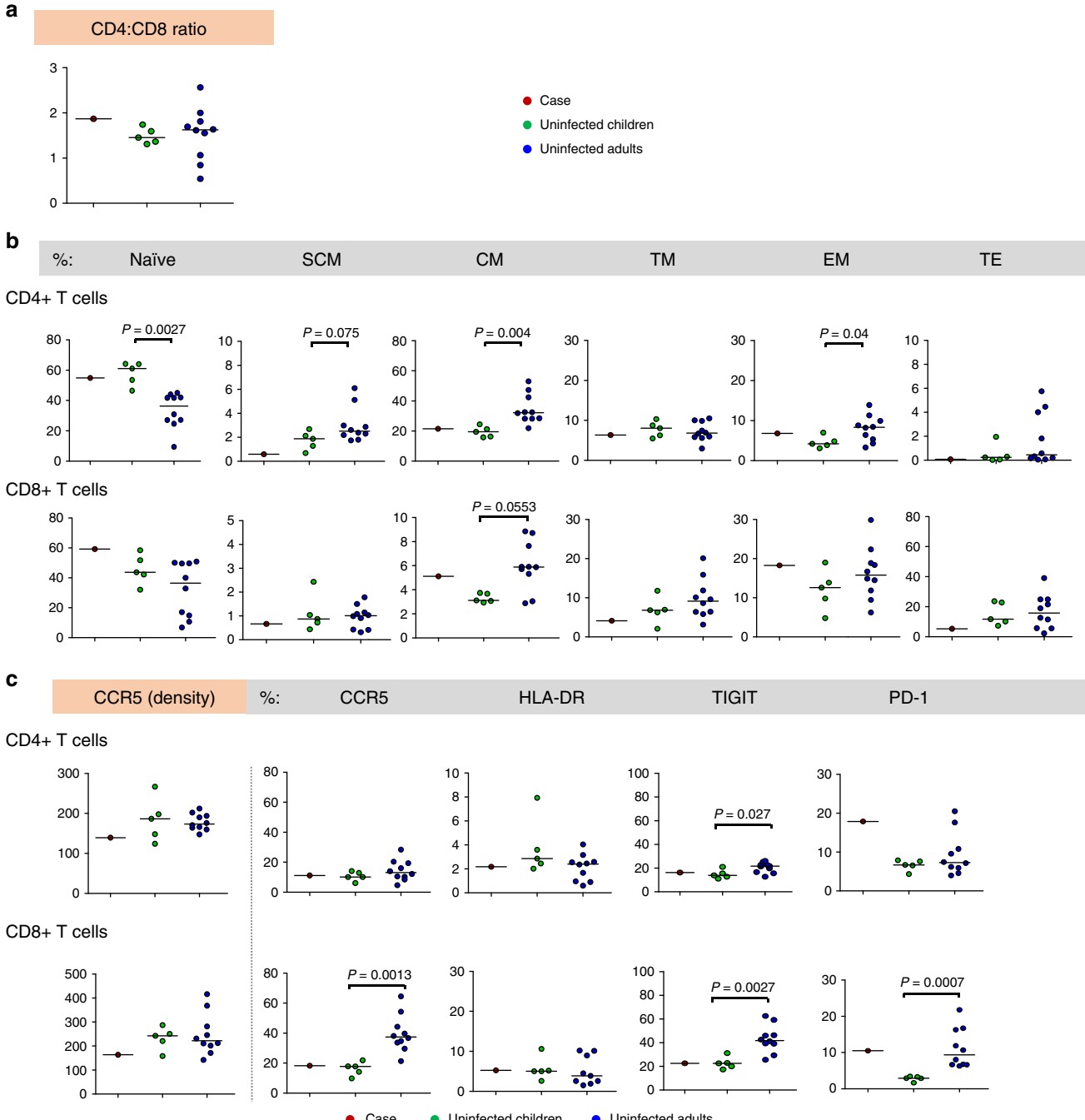

**Fig. 5** Immunophenotyping of CD4+ and CD8+ T cells. **a–c** Cell parameters determined by flow cytometry for the child at 9.5 years are compared with those of uninfected age-matched children ($n = 5$) and uninfected adults ($n = 10$): **a** CD4:CD8 ratio. **b** Percentages of CD4+ and CD8+ T cell subsets according to stage of differentiation. Naive (CD45RO−CCR7+CD62L+CD95−); SCM: stem cell memory (CD45RO−CCR7+CD62L+CD95+); CM: central memory (CD45RO+CCR7+CD62L+CD95+); TM: transitional memory (CD45RO+CCR7−CD62L+CD95+), EM: effector memory (CD45RO +CCR7−CD62L−CD95+); TE: terminal effector (CD45RO−CCR7−CD62L−CD95+). **c** CCR5 density and percentages of cells expressing CCR5, HLA-DR (immune activation), TIGIT and PD-1 (immune checkpoint inhibitory molecules) on CD4+ and CD8+ T cells. Uninfected children were compared to adults using two-tailed non-parametric Mann–Whitney U tests; individual data points are shown as coloured circles; medians are shown as solid lines; statistical significance at $P < 0.05$ is indicated. One adult outlier (extreme) data point was omitted from the CD8+HLA-DR+ graph; the adult vs. children comparison was not statistically different with or without this data point ($P > 0.05$)

An in-house ultrasensitive n-qPCR, targeting the highly conserved integrase (IN) gene designed using all subtype C sequences available on the Los Alamos HIV Database (https://www.hiv.lanl.gov/), was used to detect subtype C HIV-1 DNA and RNA. Briefly, the n-qPCR assay is a two-step PCR with the first PCR being a standard PCR allowed to proceed to end-point (30 cycles) using the Roche Expand HiFi PCR system and the second PCR a hydrolysis probe real-time PCR using Roche LightCycler 480 Probes Master Mix. The total product generated in the first-round PCR is used in the second round. Negative controls include both water and gDNA or RNA extracted from healthy HIV-1-uninfected donors. Positive controls include gDNA/RNA extracted from known HIV-1-infected individuals. Sequences of primers and probe are as follows, first-round PCR: n-IN-forward primer-1 5′-TGG CAG TRT TCA TTC ACA ATT TTA-3′; n-IN-reverse primer-1 5′-TCC TGT CYA CYT GCC ACA CAA TCA-3′; second-round qPCR: n-IN-forward primer-2 5′-CGG GTT TAT TAC AGR GAC AGC AGA G-3′; n-IN-reverse primer-2 5′-ACT ACT GCY CCT TCA CCT TTC CA-3′; n-IN-probe 5′−6 FAM-TTG GCT GGT CCT TTC CA-MGB-3′. gDNA was extracted using the QIAamp DNA Blood kit (Qiagen). RNA was

extracted from plasma or culture supernatant using the Qiagen QIAamp Viral RNA kit. Complementary DNA was synthesized using the Invitrogen SuperScript III First-Strand Synthesis System for RT-PCR (Thermo Fisher Scientific, MA, USA) and a combination of a gene-specific primer (reverse primer-1) and random hexamers. DNase treatment of extracted RNA was carried out using the Ambion TURBO DNA-free kit (Thermo Fisher Scientific). Cycling conditions were standard for both Expand HiFi and TaqMan hydrolysis probe qPCR.

**HIV-1 proviral *gag* sequencing.** The near full-length HIV-1 *gag* gene encompassing partial p17$^{Gag}$ coding regions to the Gag stop codon was PCR-amplified from gDNA isolated from CD4+ T cells using an in-house nested PCR assay. A first-round PCR amplification was performed using outer PCR primers gagoutF (5′-TGT TAA AAC ACT TAG TAT GGG CAA G-3′) and gagoutR (5′-TTA CTT TGA TAA AAC CTC CAA TTC C-3′) using the Roche FastStart HiFi PCR system (Roche Diagnostics). A second-round PCR amplification was performed using a tenth volume of first-round PCR product and second-round PCR primers gaginF (5′-TTG CAC TTA ACC CTG GCC TTT TAG A) and gaginR (5′-ATT TAT TTC TTC TAA TAC TGT ATC ATC TGC-3′). PCR products were purified using Agencourt Ampure XP magnetic bead separation (Becton-Dickinson, Franklin lakes, NJ, USA), cloned into pCR4-TOPO plasmid vectors (Invitrogen, Carlsbad, CA, USA) and transformed into *Escherichia coli* bacteria. Plasmid inserts were sequenced using BigDye Terminator v3.1 Cycle Sequencing Master Mix (Life Technologies, Carlsbad, CA, USA) and eight sequence primers. The two inner PCR primers and six overlapping internal sequencing primers (gagsegF1: 5′-GCT CTT CAG ACA GGA ACA GA-3′, gagsegF2: 5′-GGA CAT CAA GCA GCC ATG C A-3′; gagsegF3: 5′-GAA GAA ATG ATG ACA GCA TG-3′; gagsegR1: 5′-CCT GCT ATG TCA CTT CCC CT-3′; gagsegR2: 5′-TTT CCA CAT TTC CAA CAG CC-3′; gagsegR3: 5′-TTT CCA CAT TTC CAA CAG CC-3′) were used to generate *gag* sequences. Sequence products were resolved on an ABI3100 PRISM Genetic Analyser instrument (Life Technologies, Carlsbad, CA, USA). DNA sequences were analysed using Sequencher v4.5 software (Gene Codes Corporation, Ann Arbor, MI, USA). A consensus sequence (Supplementary Fig. 1) was derived from sequencing multiple cloned *gag* fragments (Genbank accession numbers: MH789553–MH789572), and viral subtype was determined by phylogenetic analysis with reference subtypes A–D using MEGA version 4[62].

**Viral outgrowth assays.** The presence of replication-competent virus was assessed through two viral outgrowth assays with modifications[63]. CD4+ T cells isolated from a stored sample of the case at 50 weeks of age ($2 \times 10^6$ cells) and from a fresh sample at 9.5 years ($2 \times 10^6$ cells) were activated with phytohemagglutinin (PHA) (5 μg mL$^{-1}$) for 2 days in media supplemented with IL-2 (20IUmL$^{-1}$; Roche Diagnostics), washed and added to PHA-stimulated CD8-depleted PBMCs ($4.0 \times 10^6$ cells) from a healthy HIV-1-uninfected donor (with high CCR5 expression as determined by flow cytometry) or $3 \times 10^5$ MOLT4/CCR5 cells. Culture supernatants were tested for HIV-1 p24 by ELISA (Alliance kit, Perkin Elmer Life Sciences Inc., UK). For increased sensitivity, the n-qPCR was used to detect HIV-1 RNA in pooled and pelleted supernatant samples. MOLT4/CCR5 cells (Cat. No. 4984) were obtained through the NIH AIDS Reagent Programme, NAIAD, NIH from Dr. Masanori Baba, Dr. Hiroshi Miyake and Dr. Yuji Iizawa[64].

**Infection of CD4 T cells in vitro.** The child's CD4+ T cells were tested for permissiveness to infection by HIV-1 BaL. CD4+ T cells ($1 \times 10^6$ cells) from the child were activated with PHA (5 μg mL$^{-1}$) in media supplemented with IL-2 (20 IU mL$^{-1}$) for 2 days, and then inoculated with HIV-1 BaL (~10 ng p24; CCR5-tropic virus; 2 h, 37 °C CO$_2$ incubator). PHA-stimulated CD8-depleted PBMCs from a healthy HIV-1-uninfected donor (high CCR5 expression) as well as $3 \times 10^5$ MOLT4/CCR5 cells infected with ~10 ng p24 HIV-1 BaL were included as positive controls. Supernatants were harvested and quantitated by p24 ELISA. HIV-1 BaL was from Dr. Suzanne Gartner, Dr. Mikulas Popovic and Dr. Robert Gallo[65] (Cat. No. 510, NIH AIDS Reagent Programme, Division of AIDS, NIAID, NIH).

**Measurement of HIV-specific antibodies.** The presence of HIV-1-specific antibodies in plasma was determined by GS HIV-1 western Blot (Bio-Rad Laboratories, Inc., USA), by ELISA (Genescreen Ultra Ag/Ab; Bio-Rad Laboratories) and a microsphere-based array assay (Bio-Rad). HIV-1 antigens were covalently conjugated onto carboxylated magnetic fluorescent beads (Bio-Plex$^{TM}$ Magnetic COOH beads; $1.25 \times 10^7$ beads per mL; Bio-Rad) using a standard two-step carbodiimide coupling procedure (Luminex MAP® technology[66]). Briefly, the stock suspension of beads was vortexed at high speed to disperse bead aggregates and an aliquot of beads ($1.25 \times 10^6$) was transferred to a coupling reaction tube, washed with dH$_2$O and resuspended in activation buffer (0.1 M NaH$_2$PO$_4$, pH 6.2), followed by the addition of 50 mg mL$^{-1}$ of *N*-hydroxysulfosuccinimide (Pierce, Rockford, IL, USA) and 1-ethyl-3-(3-dimethylaminopropyl)-carbodiimide hydrochloride (Sigma-Aldrich, St. Louis, MO, USA) and incubated for 20 min in the dark at room temperature. Activated beads were washed with coupling buffer (50 mM MES, pH 5.0) and resuspended with antigen (5–6.25 μg) to a final volume of 500 μL with 50 mM MES, pH 5.0, for 2 h, in the dark at room temperature. After incubation, the beads were washed twice with storage/blocking buffer (PBS-TBN: phosphate-buffered saline (PBS), 0.1% bovine serum albumin (BSA), 0.02%

Tween-20, 0.05% sodium azide), counted using a haemocytometer and stored in the dark at 4 °C.

HIV-specific immunoglobulin isotypes were detected by preparing a microsphere mixture comprising the seven HIV antigen-coupled beads (gp41, Gag, RT, gp120, Nef, Tat, Vpu, peptide Vpu9 and V1V2 scaffold; Supplementary Table 4). To each well of a 96-well flat bottom Greiner plate (Bio-Rad), 50 μL of working microsphere mixture prepared in assay buffer (PBS, 1% BSA; 2500 beads from each bead region per well), 40 μL assay buffer and 25 μL of patient plasma (diluted 1:75 in assay buffer) was added. Following an incubation period (on a shaker in the dark at room temperature; 2 h), the plate was washed three times with wash buffer (PBS, 0.1% BSA, 0.05% Tween-20) with a final wash in assay buffer. HIV-specific antibody isotypes were detected by adding 50 μL per well at 2 μg mL$^{-1}$, *R*-phycoerythrin-conjugated mouse anti-human IgG$_1$ to IgG$_4$ (Cat. Nos 9052-09, 9070-09, 9210-09, 9200-09, respectively, Southern Biotech, USA), mouse anti-human IgM (Cat. No. 9020-09, Southern Biotech), mouse anti-human IgA$_1$ (Cat. No. 9130-09, Southern Biotech) or mouse anti-human IgA$_2$ (Cat. No. 9140-09, Southern Biotech) with shaking (in the dark at room temperature) followed by three washes. Beads were finally resuspended in Bio-Rad sheath fluid and analysed using the Bio-Plex 200 instrument (Bio-Rad) by measuring the fluorescence signal for 50 beads per analyte. Background signal, defined as the mean fluorescence intensity (MFI) for each microsphere set when incubated with PE detection Ab in the absence of patient plasma, is subtracted from the fluorescent intensity of each sample. The cut-off for seropositivity for each analyte is calculated as the average MFI value from pooled plasma from HIV-1-uninfected patients plus 3 standard deviations. HIV-Ig (Cat. No. 3957; NIH AIDS Reagent Programme) served as a positive control on each plate and the MFI values tracked with a Levy–Jennings plot.

**Intracellular cytokine assays.** HIV-1-specific T cell (CD4+, CD8+) and NK cell responses were measured using a whole blood intracellular cytokine assay (IFN-γ and IL-2) stimulated with overlapping HIV-1 subtype C synthetic peptides in pools representing Gag, Pol, Nef, Env, Tat, Rev, Vif, Vpu and Vpr proteins. Briefly, 200 μL of whole blood, collected in sodium heparin, was stimulated with a final concentration of 10 μg mL$^{-1}$ of peptide (synthesized as 15–18 mers overlapping by 10 amino acids, with the exception of Nef, which overlaps by 11 amino acids, NMI, Germany) together with 1 μg of the costimulatory antibodies CD28 and CD49d (Cat. Nos 340975 and 340976, respectively, BD BioSciences) and the transport inhibitor brefeldin A (10 μg mL$^{-1}$; Sigma-Aldrich). Positive controls were stimulated with 1 μg/mL SEB (Cat. No. S4881, Sigma-Aldrich) for T cells or with 5 ng mL$^{-1}$ anti-CD16 antibody (Clone eBioCB16, Cat. No. 16-0168-85, eBioscience, San Diego, CA, USA) for NK cells. The latter measures reverse ADCC, therefore providing a measure of the capacity of NK cells to perform ADCC through engagement with FcγRIIIa/CD16a receptors. To monitor spontaneous cytokine release, the equivalent amount of dimethyl sulfoxide, as in the peptide tubes, together with the costimulatory antibodies was prepared. Samples were incubated for 6 h at 37 °C, after which they were maintained at 18 °C overnight. Twenty microlitres of ethylenediaminetetraacetic acid (EDTA) was added for 15 min, following which red blood cells were lysed for 10 min at room temperature using 2 mL FACS lysing solution (BD Biosciences). FACS permeabilizing solution (500 μL; 15 min; room temperature; BD Biosciences) was added to centrifuged samples which were washed twice before staining with CD3 PerCP (clone SK7, Cat. No. 345766, 5 μL), CD8 FITC (clone SK1, Cat. No. 347313, 5 μL), CD4 BV786 (clone L200, Cat. No. 563914, 0.8 μL), CD56 APC (clone NCAM16.2, Cat. No. 341025, 5 μL of a 1:16 dilution), IL-2 phycoerythrin (PE) (clone 5344.111, Cat. No. 340450, 10 μL) and IFN-γ PE (clone 25723.11, Cat. No. 340452, 5 μL) for 60 min at room temperature in the dark. Stained cells were acquired using a 4-laser BD LSRFortessa$^{TM}$ X-20 flow cytometer (BD Biosciences) and analysed (gating strategy in Supplementary Fig. 2) using the FlowJo Software (Tree Star Inc., Ashland, OR, USA).

**Multiparameter flow cytometry.** Whole blood immunophenotyping by multi-colour flow cytometry (to determine proportions of naive/memory CD4+ and CD8+ T cell subsets) included measures of CCR5 expression (density and %), immune activation (HLA-DR) and immune exhaustion (TIGIT and PD-1). The following antibodies were used, in three antibody panels: CD3 APC-H7 (clone SK7, Cat. No. 560176, 2 μL), CD8 PerCP (clone SK1, Cat. No. 347314, 6.5 μL), CD8 Alexa Fluor 700 (clone RPA-T8, Cat. No. 557945, 2 μL), CD4 BV786 (clone L200, Cat. No. 563914, 0.8 μL), CD4 FITC (clone SK3, Cat. No. 347413, 6 μL), CCR5 PE (clone 2D7, custom 1:1 conjugated antibody, 10 μL), CCR7 FITC (clone 150503, Cat. No. 150503, 6.5 μL), CD45RO BV510 (clone UCHL1, Cat. No. 563215, 4 μL), CD62L PE-CF594 (clone DREG-56, 1.6 μL), PD-1 BV786 (clone EH12.1, 3.3 μL) were obtained from BD Biosciences. TIGIT APC (MBSA43, Cat. No. 562301, 1.6 μL) was obtained from eBioscience, HLA-DR PE-Cy5.5 (clone TU36, Cat. No. MHLDR18, 1.6 μL) was obtained from Life Technologies and CD95 BV605 (clone DX2, Cat. No. 305628, 6.5 μL) was obtained from BioLegend (San Diego, CA, USA). The CCR5 antibody was conjugated to PE at a ratio of 1:1, and quantitation (density measured as antibodies bound per cell) was carried out using the QuantiBRITE system (BD Biosciences). Whole blood (100 μL) was incubated with the antibodies at room temperature, in the dark, for 15 min. Thereafter, red blood cells were lysed using FACS lysing solution (BD Biosciences). Samples were then washed and resuspended in FACSflow. Stained cells were acquired on a 4-laser BD LSRFortessa$^{TM}$ X-20 flow cytometer (BD Biosciences) and analysed (antibody panels and

gating strategies in Supplementary Fig. 3-5) using the FlowJo Software (Tree Star Inc.).

**Host genotyping.** *HLA* class I and II and *KIR* genotyping was conducted on gDNA extracted from blood cells (QiaAmp DNA blood mini kit, Qiagen). The sequence-based typing resolver kits (Conexio Genomics, Fremantle, Australia) were used to generate HLA-A, HLA-B, HLA-C, DRB1, DPB1 and DQB1 amplicons for DNA sequencing as described by the manufacturers. The exons included for DNA sequencing are: exons 1–4 for HLA-A and HLA-B; exons 1–8 for HLA-C; exons 2 and 3 for HLA-DRB1 and HLA-DQB1; exons 1–5 for HLA-DPB1.

The presence or absence of the 16 *KIR* genes (14 functional and 2 pseudogenes *2DP1* and *3DP1*) were determined using allele-specific (AS) primers (Supplementary Table 5) in a real-time PCR assay[67]. Briefly, PCR reactions were performed in a 5 μL volume, containing 2× Maxima SYBR Green/ROX qPCR Master Mix (Fermentas, Burlington, ON, Canada), 0.2 μM of *KIR*-specific primers, 0.2 μM of *galactosylceramidase*-specific primers and 5 ng of DNA. Thermocycling was performed using the Applied Biosystems 7500 Real-Time PCR system (Applied Biosystems, Foster City, CA, USA) under the following conditions: 95 °C for 10 min, followed by 30 cycles of 95 °C for 15 s and 60 °C for 1 min, with subsequent melt-curve analysis.

Published primers and probe specific for *KIR2DS4* full-length (*f*; *\*001*) or truncated (*v*; *\*003,004,006,007,009*) alleles[68] and *KIR3DL1/S1*[69] were used in combination with the primers and probe specific for the human β-globin (*BGB*) reference gene[70] (Supplementary Table 6) in a probe hydrolysis-based relative quantification real-time allele-specific (AS)-PCR assay to determine gene copy number variation (*KIR2DS4* and *KIR3DL1/3DS1*). To facilitate target and reference gene multiplexing, *KIR*-specific probes were labelled at the 5′ end with the fluorochrome VIC, while *BGB* probes were labelled with the fluorochrome FAM. Control samples of known gene copy numbers were run concurrently with unknowns and gene copy numbers inferred using a delta Ct method (difference between Ct values obtained for *KIR* and *BGB* genes). All samples were run in duplicate. Real-time AS-PCR amplification was performed in 96-microwell PCR plates using an ABI7500 real-time PCR instrument (Life Technologies, Carlsbad, CA, USA). Reaction volumes were 5 μL containing 5 ng of genomic DNA, 2× LightCycler 480 Probes Master Mix (Roche), 0.5 μM *KIR3DL1/S1* or *KIR2DS4f/v* forward/reverse primers (Inqaba Biotec, Pretoria, RSA), 0.5 μM *BGB* forward/ reverse primers (Inqaba Biotec, Pretoria, RSA), 0.1 μM VIC-labelled *KIR3DL1/S1* or *KIR2DS4f/v* probe and 0.1 μM FAM-labelled *BGB* probe (Life Technologies, Carlsbad, CA, USA). Cycling conditions were an initial incubation of 95 °C for 10 min, followed by 40 cycles of 95 °C for 15 s, 55 °C for 10 s and 60 °C for 30 s.

## Data availability

The data that support the findings in this study are available from the corresponding authors upon reasonable request. The source data underlying Figs 1a–c and 5a–c are provided in Supplementary Tables 1 and 3, respectively. Genbank accession numbers for cloned *gag* proviral sequences: MH789553–MH789572.

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

## Acknowledgements

The research was supported by the South African Research Chairs Initiative of the Department of Science and Technology and National Research Foundation, the South African Medical Research Council with funds received from the South African National Department of Health, National Institutes of Allergy and Infectious Diseases (Comprehensive International Programme for Research on AIDS (CIPRA)–South Africa (U19 AI153217), Departments of Health of the Western Cape and Gauteng, South Africa, ViiV Healthcare, National Institute of Mental Health (T32 MH19105-28), the Eunice Kennedy Shriver National Institute of Child Health and Human Development (RO1 HD073952) and the EPIICAL project (http://www.epiical.org/) funded through an independent grant by ViiV Healthcare UK.

## Author contributions

A.V. and C.T.T. conceived, designed and obtained ethical clearance for the study. A.V., M.F.C, L.K., A.L., J.M., A.B. and D.G. contributed the clinical data. K.O. analysed the clinical data. M.P. performed and analysed data from sensitive HIV RNA and DNA assays. D.B.S. performed and analysed data from viral growth assays and from HIV-specific antibody profiling. D.B.S. and S.S. performed and analysed data from intracellular cytokine assays, respectively. S.S. developed the immunophenotyping flow cytometry panels. B.DC.D. and S.S. conducted immunophenotyping assays and analysed the flow data. S.L. performed host genotyping and virus subtyping. C.T.T. interpreted the collective scientific data and wrote the manuscript, which was reviewed and edited by all authors.

## Additional information

**Competing interests:** The authors declare no competing interests.

