## [Peer Review File · Nature Communications]

Reviewers' Comments:

Reviewer #1:

Remarks to the Author:

This report by Violari and colleagues describe the case of a perinatally HIV-infected child who durably controls viremia after treatment interruption. Although interesting, this is not novel as several cases of transient or durable post-treatment control have been reported both in adults (Lodi et al ArchInt Med 2012; Saez-Cirion et al PlosPath 2013; Kinloch-de Loes OFID 2015; Perkins et al OFID 2015; McMahon et al AIDS 2017; Maggiolo et al AIDS 2018) and children (Persaud et al NEJM 2013; Frange et al Lancet HIV 2015) in the last few years.

The authors have analysed several parameters to characterize the case but, because the results can only be considered descriptive, the discussion is too speculative in occasions, in particular regarding the persistence of replication competent virus in this child.

For instance the authors mention that "Lack of CD8+ T-cell response suggests that CD8+ T-cells are not currently involved in controlling levels of viraemia, supporting the possibility that HIV may be defective for further infection of permissive cells". There are multiple possibilities to explain lack/weak CD8 T cell responses. Responses are generally very weak or undetectable in infants treated very early and would remain so if the virus is well controlled. Very weak CD8+ T cell responses characterize elite controllers showing a very tight control of infection for several years, despite presence of replication competent virus. The viral outgrowth assays performed to detect replication competent virus are insufficient to conclude. The number of cells tested (2M cells) is very limited . It is unclear whether fresh or frozen cell samples were used, the latter would decrease the sensibility of the assay. In any case, replication competent virus has been shown to persist in individuals in whom much larger amount of cells were tested in qVOA assays that provide negative results (e.g. 22M cells in the case of the Mississippi baby).

It would be interesting to show antibodies, T cell responses and T cell phenotype compared to other early treated children, and not only to non-infected individuals.

Did the authors test for the presence of antiretroviral drugs during the control period?

Reviewer #2:

Remarks to the Author:

Summary

The authors describe a fascinating case of post-treatment control of HIV in a perinatally infected child. The virologic and immunological features are presented and their possible contribution to the outcome discussed.

General comments:

This is an extremely interesting case, and one that deserves the highest level of scrutiny since this child is one of only 3 described to have achieved cure/remission. Defining the mechanisms by which post-treatment control can be achieved is of critical importance. The authors provide a comprehensive analysis of the relevant virology and immunology. An anecdotal case on its own cannot definitively define mechanism but this case adds to the other rare instances of post-treatment control in paediatric infection in providing important clues that will help advance the field.

Specific comments

1. Access to samples of PBMC and/or plasma from the two timepoints prior to ART initiation (at 6 and 8 wks) and at the time of ART cessation (at 50 and 58 wks) clearly would be useful to help explore factors contributing to the outcome in the child. Presumably there are none available from the pre-treatment timepoints – if so, it would be helpful to clarify this. If not, it should be possible to characterise the virus to better address the possibility raised in the Discussion that the virus might be defective. For example full-length sequencing would be helpful in identifying deletions or mutations that might significantly reduce viral replicative capacity. This could be further tested in fitness assays. In addition, comparison of the virus at 6wks and 8wks, at the time of a dramatic and highly unusual drop in viral load, might provide clues as to the mechanism involved. Comparison of viral sequence at these pre-ART timepoints with the sequence the authors have constructed at 50 weeks might also provide clues. I am sure the authors have considered these but do not have the samples, but it would be helpful to clarify if so.

2. The second striking aspect about this case (the first being the drop in viral load between 6 and 8 wks) is the rise in CD4% to 65% or thereabouts immediately post treatment cessation (58 wks). It is surprising that this neither received any comment nor was investigated. It is possible that this was an artefact, since such a leap in CD4% is so remarkable. However, given that this is a unique child, it is also possible that the CD4 T-cell response may have played a part in immune control of HIV post treatment interruption. With this in mind, it is noteworthy that the authors describe a detectable Gag-specific CD4+ T-cell response but no HIV-specific CD8+ T-cell response. Although this combination of responses is unusual, a dominant HIV-specific CD4 response is certainly well-described among adult elite controllers. It is possible that in this child, following post-treatment control, HIV-specific CD8+ T-cell responses are undetectable but a low-level Gag-specific CD4+ T-cell response remains. It would be interesting to investigate whether a large Gag-specific CD4+ T-cell response is responsible for the massive CD4 expansion at the 58 week timepoint, if this is not an artefact and if cells are available. In addition to IL-2/IFN-g responses it would be illuminating to look more broadly at HIV-specific CD4 T cell functions eg cytotoxicity (Soghoian et al Sci Transl Med), proliferative activity, etc.

3. The Discussion draws attention to a number of features observed in the child, immunological or genetic, that have previously been associated with immune control. Given that many of these arise in a high proportion of subjects and individually have quite a modest impact on control of viraemia, one might equally reach the opposite conclusion than that reached by the authors, ie that the immunology and genetics presented in the child are not exceptional. This, indeed, is what has been one of the key findings with respect to HLA type in post-treatment control. It may be that the potential significance of the factors discussed by the authors could be downplayed since there is no clear-cut factor explaining post-treatment control in this child.

Minor comments

1. The authors state that the NK response in the child is 'very strong' and that the SEB responses are 'strong' (Fig 3C). I am not sure how useful these data are in isolation.

2. To use birth weight as a discriminator between intra partum and intrauterine infection seems a stretch (para 2 of the Discussion). It would be more accurate to clarify that the timing of infection in this child is unknown. The Mississippi baby had a normal birth weight for 35 weeks' gestation.

3. It would be helpful to clarify that no sample was available from the mother that would have confirmed mother-to-child transmission. Also, there is no mention of the mother, perhaps because there is no information, but are there any CD4 or viral load data? The mother of the Mississippi baby had a healthy CD4 count (644/mm³) and a low viral load (~2,000 c/ml).

4. It may be helpful to show this child's initial CD4, CD4% and viral load in the context of the other 226 CHER children who received early ART. This would show where the CD4% in this child is among the group at 8 wks' age: 41.6% sounds very high but readers may not be aware that an entry criterion was a CD4% of >25%. In addition, the comparisons shown with age-matched children and

adults, whilst of interest, would be more telling if they could be made with others in the CHER cohort prior to ART interruption. However I imagine the samples are not available for such comparisons.
5. Finally, the sex of the child may have some bearing on the outcome and there is no mention of this. It would be helpful to explain that permission for this information to be included was not provided in this case.

Reviewer - Philip Goulder

Reviewer #3:

Remarks to the Author:

The authors report on a child that was perinatally infected with HIV-1 and received early limited (40 weeks) antiretroviral Treatment as part of the children with early antiRetroviral therapy (CHER) Trial. At Age 9.5 years, diagnostic tests for HIV were negative. Characteristics such as CD4:CD8 Ratio, low T-cell activation and low CCR5 Expression were noted, resembling those of uninfected children. Virus persisted (HIV-1 DNA and Plasma RNA) as measured by sensitive assays but replication-competent Virus was not detected.

This is a very interesting case of a posttreatment Controller. Those patients are rarely identified and have given rise to many studies that want to identify factors explaining posttreatment control.

Although the authors have undertaken many attempts to identify factors explaining this very rare phenomenon, they cannot really do this as in most case reports this is intrinsically difficult with a case of one and due to the heterogeneity of all These cases, large studies will probably never be possible. What is interesting is that the child's virus before treatment replicated a lot and thus was clearly replication competent at that time. Maybe they could also discuss the very recent paper by Colby et al, Nat Med, 2018 Jun 11. doi: 10.1038/s41591-018-0026-6, which did not find any posttreatment control in Fiebig 1 and 2 treated adult patients. Thus, it seems that some replication of the virus is needed for posttreatment control.

It also seems to be clear that cellular adaptive immune response is not the major reason for containing the virus. Neutralizing activity of the plasma was not tested as far as I can infer from the manuscript but most likely that was also not the case because neutralizing antibodies rarely seem to be of importance in these post treatment controllers and time for eliciting bNabs before Treatment was too short anyway. An interesting finding is the very low expression of CCR5. Host genetic factors as measured do not explain much.

Taken together, this is an interesting case report, which however, still leaves open, why this is a posttreatment Controller.

What would be interesting to look at further is:

- 1) Full length genome sequencing of the virus (might be difficult to get at these low copy numbers). This would probably be the most interesting additional test. The big question is, whether the Virus is really replication competent or not. It is interesting that Plasma RNA is still found at very low Levels, thus, Virus particles must be produced (not just transcripts because they would be degraded rapidly in the Plasma). Could there be a possibility that there are Long-lived cells producing viruses with replication defective particles?
- 2) exome sequencing of the host (maybe there is some very rare variant of relevance to be found)
- 3) cellular HIV-RNA (at least unspliced)
- 4) neutralizing antibody Responses if possible

Minor comments:

Ref 1, should be completed by Finzi, et al, 1997, Science and Wong et al, Science.

Manuscript: NCOMMS-18-14830

Title: **A child with perinatal HIV infection and long-term sustained virological control following antiretroviral treatment cessation**

Reviewers' comments:

Reviewer #1 (Remarks to the Author):

1. This report by Violari and colleagues describe the case of a perinatally HIV-infected child who durably controls viremia after treatment interruption. Although interesting, this is not novel as several cases of transient or durable post-treatment control have been reported both in adults (Lodi et al Arch Int Med 2012; Saez-Cirion et al PlosPath 2013; Kinloch-de Loes OFID 2015; Perkins et al OFID 2015; McMahon et al AIDS 2017; Maggiolo et al AIDS 2018) and children (Persaud et al NEJM 2013; Frange et al Lancet HIV 2015) in the last few years.

Response:

We thank the reviewer for the additional references. We do think that our report of an HIV-infected male child in remission is novel in itself. The phenotype of post-treatment control is more readily shown in adults as outlined by the reviewer. However, we do think this child is unique given that 226 children from the CHER trial, who received early treatment did not achieve this outcome (1/227 = 0.4%; 1/178 virally suppressed at interruption: 0.6%). This highlights a rare occurrence. The French case was part of the French paediatric cohort (n=173) of early treated children, but was one of only 15 children who stopped ART with an undetectable viral load (6.7%). These are the only denominators available for children.

Importantly this child comes from a region most affected by the HIV epidemic where achieving remission might be considered more challenging and is not yet reported. We feel the features associated with this child are of interest, many of which have not been described in other cases of post-treatment control, some of whom began ART during acute infection (Perkins, Kinloch-de Loes, Lodi and others during chronic infection (MacMahon, Maggiolo). Unlike the Mississippi child, who had short term remission, our patient has sustained remission for 8.5 years. As there are differences to the adolescent reported by Frange et al, our case report contributes to documenting the heterogeneity of responses. We feel that our extensive preliminary investigations contribute to creating a profile for post-treatment controllers and are hypothesis generating for further study in adult and children cohorts.

Changes to manuscript: We added a paragraph to the introduction to include the additional references mentioned by the reviewer and highlighted the key paediatric cases of early treatment. We have included more description of the similarities and differences (also in the 'discussion'), especially with the French case – which is most like our child in timing of treatment and long term sustained suppression off ART.

To not overstate the uniqueness of this child - we have removed “unique” from the title and reworded to improve the grammar. However, we have retained the use of “unique” in the abstract.

2. The authors have analysed several parameters to characterize the case but, because the results can only be considered descriptive, the discussion is too speculative in occasions, in particular regarding the persistence of replication competent virus in this child.

For instance the authors mention that "Lack of CD8+ T-cell response suggests that CD8+ T-cells are not currently involved in controlling levels of viraemia, supporting the possibility that HIV may be defective for further infection of permissive cells". There are multiple possibilities to explain lack/weak CD8 T cell responses. Responses are generally very weak or undetectable in infants treated very early and would remain so if the virus is well controlled. Very weak CD8+ T cell responses characterize elite controllers showing a very tight control of infection for several years, despite presence of replication competent virus. The viral outgrowth assays performed to detect replication competent virus are insufficient to conclude. The number of cells tested (2M cells) is very limited. It is unclear whether fresh or frozen cell samples were used, the latter would decrease the sensitivity of the assay. In any case, replication-competent virus has been shown to persist in individuals in whom much larger amount of cells were tested in qVOA assays that provide negative results (e.g. 22M cells in the case of the Mississippi baby).

Response:

We feel that our arguments for the possible state of the virus at 9.5 years of age are logical and allow us to derive a working hypothesis that we intend addressing in detail in future studies. Below we address the reviewer's concerns.

The CD8+ T cell response:

In our discussion, we state that lack of CD8+ T cell responses suggests their not currently being involved in controlling viraemia, therefore supporting the possibility of HIV being defective for further replication. This is very plausible for the following reasons: if CD8+ T cells were being constantly challenged by HIV then one would expect these responses to be readily detected. We also noted that it is remarkable to have a CD4+ T cell response yet no CD8+ T cell response. In an HIV-1 infected individual one seldom sees an HIV-specific CD4+ T cell response that is greater than a CD8+ T cell response. This corresponds with the expected expansion of CD8+ T cells in response to viral challenge relative to the loss of CD4+ T cells, or even if CD4 counts are maintained at healthy levels. This is also normally accompanied by increased activation of these cells. CD4 T cells could be primed by defective virus as these cells respond to exogenous proteins, however CD8 T cells require endogenous processing of virus within infected cells so would be less likely be detected or maintained if new cells were not being infected (or to a very low extent). We appreciate that CD8+ T cell responses may be below the level of detection in our assay (as one may expect there would be memory of earlier encounter with virus).

The Reviewer also refers to very weak CD8+ T cell responses in some elite controllers with very tight control of viraemia, despite replication-competent virus being present. There is heterogeneity even among elite controllers, grouped according to their ability to control viraemia in the absence of ART. Within this group are those with very substantial CD8+ T cell responses (broad and of high magnitude) and those referred to by the reviewer with weaker responses. In those with broad responses of high magnitude, the response is likely maintained by ongoing exposure to HIV. For those with low responses, it is very likely that other mechanisms contribute to HIV control even if weaker CD8+ T cells are playing a role. Elite controllers also have increased immune activation, particularly evident on CD8+ T cells. Our case has neither detectable CD8+ T cell responses nor increased immune activation, but rather a demonstrable CD4 Gag response. We had added that such responses - CD4 but no CD8 response - has been shown in exposed uninfected individuals, but only using much more sensitive methods – making our detection of the CD4 response even more remarkable.

We have mentioned that lack of HIV-specific T cell responses could be a result of early treatment. However, a weak but demonstrable CD4+ T cell response is present in the absence of a CD8+ T cell response, which we have highlighted. We would expect that drugs would diminish the possibility of both CD4+ and CD8+ T cell responses.

The manuscript discusses these unusual presentations of T cell responses in paragraphs 6 and 7 of the 'discussion'. We changed the sentence slightly to be more cautious in suggesting that the lack of a CD8+ T cell response supports the defective virus theory – "...might not be currently involved in controlling levels of viraemia, supporting the possibility. ...". We have also considered that such a response could have been (and very likely was) operational early in life.

Testing of 2 million CD4 T cells – limited in number:

We have stated in the discussion that "Inability to detect replication-competent virus *in vitro* may be because of assay sensitivity (2 million CD4+ cells tested), or defective virus that cannot accumulate to detectable levels." Therefore, the findings should be viewed under these testing conditions.

We used fresh material at 9.5 years of age, now included in the 'methods' section. We also used cells from a high CCR5-expressing donor to maximize our chances of detecting replication-competent virus. However, our sample from week 50 was frozen. In the French case, the conditions for testing for replication-competent virus were identical to this case. The authors also tested 2 million CD4+ T cells and used CD8+-depleted PBMCs, readily detecting replication-competent virus.

We have not stated categorically that there is no replication-competent virus in the child. However, we do know that although the virus persists, viral rebound has not occurred. For the Mississippi child, despite not finding replication competent virus in 22 million resting CD4+ T cells (a very large number to get from a child) at 24 months of age, (Persaud et al 2013), the true test of persistence was revealed when virus rebounded after 27 months off ART (Luzuriaga et al, 2015). The case of the Mississippi baby highlights that even when no replication competent virus is detected after using a large number of cells, a rebound may still occur. This is further supported by cases of very early treated adults, in whom despite exhaustive tests to show lack of viral persistence, virus rebounded shortly after treatment interruption.

To better clarify these comparisons, we modified paragraph 2 of the 'discussion' to highlight differences and similarities between cases, and have also added the 2 recent reports of very early adult treatment of (Henrich et al, 2017, Colby et al, 2017). This highlights that we do not have adequate biomarkers that predict likelihood of rebound if ART is stopped, and that some limited viral replication may be necessary for achieving long term control of viraemia.

The French case also showed weak T cell responses to Gag, Pol and Nef – we have added this to paragraph 6 of the 'discussion'.

3. It would be interesting to show antibodies, T cell responses and T cell phenotype compared to other early treated children, and not only to non-infected individuals.

Response:

We are interested to compare HIV antibodies, T cell responses and T cell phenotypes of the child versus other early treated children. We are planning such studies which require the recruitment of new patients. However, we consider that such studies (which, in their own right, are extensive) as beyond the scope of this initial report. We prefer rather to thoroughly evaluate this child before extending into other cohorts for further comparison.

4. Did the authors test for the presence of antiretroviral drugs during the control period?

Response:

Yes, we have now included this important point in the case description (2nd paragraph of the 'results').

Reviewer #2 (Remarks to the Author):

Summary

The authors describe a fascinating case of post-treatment control of HIV in a perinatally infected child. The virologic and immunological features are presented and their possible contribution to the outcome discussed.

General comments:

This is an extremely interesting case, and one that deserves the highest level of scrutiny since this child is one of only 3 described to have achieved cure/remission. Defining the mechanisms by which post-treatment control can be achieved is of critical importance. The authors provide a comprehensive analysis of the relevant virology and immunology. An anecdotal case on its own cannot definitively define mechanism but this case adds to the other rare instances of post-treatment control in paediatric infection in providing important clues that will help advance the field.

Response:

We appreciate the very insightful and helpful comments and suggestions from the reviewer.

Specific comments:

1. Access to samples of PBMC and/or plasma from the two timepoints prior to ART initiation (at 6 and 8 wks) and at the time of ART cessation (at 50 and 58 wks) clearly would be useful to help explore factors contributing to the outcome in the child. Presumably there are none available from the pre-treatment timepoints – if so, it would be helpful to clarify this. If not, it should be possible to characterise the virus to better address the possibility raised in the Discussion that the virus might be defective. For example full-length sequencing would be helpful in identifying deletions or mutations that might significantly reduce viral replicative capacity. This could be further tested in fitness assays. In addition, comparison of the virus at 6wks and 8wks, at the time of a dramatic and highly unusual drop in viral load, might provide clues as to the mechanism involved. Comparison of viral sequence at these pre-ART timepoints with the sequence the authors have constructed at 50 weeks might also provide clues. I am sure the authors have considered these but do not have the samples, but it would be helpful to clarify if so.

Response:

We agree that studying samples within the first year of life are crucial to understanding events leading to this outcome. However, we only have one pre-treatment sample ($\pm 10^6$ PBMCs) in storage which we are reluctant to use only for virus studies as we also want to investigate potentially important immune responses at that time point. Therefore, gleaned as much information as possible prior to this step, will inform our best use of this very important single sample. We feel very

strongly that this is the responsible thing to do, rather than in haste, to lose a very important opportunity to identify a potentially relevant immune marker associated with remission.

Our approach therefore, as suggested by Reviewer #2, is to explore full-length sequencing, firstly of cell-associated proviral DNA at 9.5 years of age – this will give us a ‘map’ of all deletions and mutations that may affect virus replication. In addition, we will gain a whole genomic view of possible HIV escape mutations in the context of the child’s specific HLA class I and II alleles. We would next move on to a “dispensable” sample at a time point close to treatment interruption (after 50 weeks) as the sample at 50 weeks is fully used. We expect this analysis may reveal that the virus (provirus) has changed little in the 8.75 years of virological control. If, however, substantial changes are noted, we will look at other intermediate time points to establish the longitudinal profile of proviral reservoir variation over time. Depending on results, we will conduct viral fitness assays. Ultimately, we hope to compare the pre- ART virus (no plasma, only one vial PBMCs available) to all the other whole genome sequences. Our findings in this area, could preclude the need to find replication-competent virus in larger numbers of CD4+ T cells.

We already know very small amounts of virus are being produced *in vivo* (6.6 RNA copies/ml). In order to obtain sufficient virus for full-length sequencing we are currently storing blood specimens from multiple visits. This will require a number of years (assuming that we still have informed consent to continue these investigations) but will be a very important comparison well worth the effort.

2. The second striking aspect about this case (the first being the drop in viral load between 6 and 8 wks) is the rise in CD4% to 65% or thereabouts immediately post treatment cessation (58 wks). It is surprising that this neither received any comment nor was investigated. It is possible that this was an artefact, since such a leap in CD4% is so remarkable. However, given that this is a unique child, it is also possible that the CD4 T-cell response may have played a part in immune control of HIV post treatment interruption. With this in mind, it is noteworthy that the authors describe a detectable Gag-specific CD4+ T-cell response but no HIV-specific CD8+ T-cell response. Although this combination of responses is unusual, a dominant HIV-specific CD4 response is certainly well-described among adult elite controllers. It is possible that in this child, following post-treatment control, HIV-specific CD8+ T-cell responses are undetectable but a low-level Gag-specific CD4+ T-cell response remains. It would be interesting to investigate whether a large Gag-specific CD4+ T-cell response is responsible for the massive CD4 expansion at the 58 week timepoint, if this is not an artefact and if cells are available. In addition to IL-2/IFN- γ responses it would be illuminating to look more broadly at HIV-specific CD4 T cell functions eg cytotoxicity (Soghoian et al Sci Transl Med), proliferative activity, etc.

Response:

Many thanks for highlighting the unusual rise in CD4% post-ART at 58 weeks of age, an important omission from our manuscript.

Upon close inspection of the CD4 and CD8 T cell data (percentages together were in excess of the CD3% which would ordinarily be flagged as problematic), it became apparent that there may have been an error in the results – either a transcription error or in the original flow cytometry analysis. We requested copies of archived results, these confirmed that there was no error in transcribing the results. Raw data of plots was then obtained and re-analyzed by flow cytometry. We concluded that the original data was incorrect (a result of high background in sample and not adequately gating

target cell populations). The correct values are as follows - CD4%: 42.7%, CD4 count: 1857. Therefore, there is no notable rise in CD4% at this time point which would require any discussion.

We redrew Figure 1 b and c with the correct values. We appreciate the opportunity to correct these values. Without the reviewer's comment, we would have overlooked this aspect.

Future work will address the fine specificity of the CD4 T cell response the child currently has (determining which viral epitopes are targeted by the CD4 T cells – using overlapping Gag peptides to map these) as well as functional attributes of these cells such as cytokine production and capacity for cytotoxic activity. Because this is a weak response, we will expand the Gag-specific cells *in vitro* to improve sensitivity. This expansion approach may also “rescue” a CD8+ T cell response to Gag that is not detectable with our whole blood assay at 9.5 years, as one would expect there would be some memory of CD8 T cell encounter with virus in early life. Sample availability and volumes at earlier time points make it unlikely we will be able to effectively study all these parameters – as mentioned earlier we would rationalize the use of any such samples very carefully.

3. The Discussion draws attention to a number of features observed in the child, immunological or genetic, that have previously been associated with immune control. Given that many of these arise in a high proportion of subjects and individually have quite a modest impact on control of viraemia, one might equally reach the opposite conclusion than that reached by the authors, ie that the immunology and genetics presented in the child are not exceptional. This, indeed, is what has been one of the key findings with respect to HLA type in post-treatment control. It may be that the potential significance of the factors discussed by the authors could be downplayed since there is no clear-cut factor explaining post-treatment control in this child.

Response:

The reviewer makes a valid point. However, different protective factors are likely to exist in different combinations in groups of controllers that could generate hypotheses to be tested.

We present our findings as possible contributing features warranting further investigation to evaluate their importance. We have stated, however, that there is a potential capacity for diverse responses (generally considered advantageous) as the host would have more “options” for engaging with virus and preventing evasion of immune responses. While we do not consider these as exceptional, perhaps some in combination might have a role in virus suppression.

We have added the contrasting result of homozygosity at 3 loci in the French case – to paragraph 8 of the ‘discussion’.

Minor comments

1. The authors state that the NK response in the child is ‘very strong’ and that the SEB responses are ‘strong’ (Fig 3C). I am not sure how useful these data are in isolation.

Response:

In legend for Figure 3, we provide ranges for healthy adults in our populations – the child has adult-like values for both these parameters. This illustrates a healthy capacity for response of these cell types when engaging in antiviral responses (also serves as a positive control for the assays, ensuring that any negative results are not due to lack of responsiveness/energy).

2. To use birth weight as a discriminator between intra partum and intrauterine infection seems a stretch (para 2 of the Discussion). It would be more accurate to clarify that the timing of infection in this child is unknown. The Mississippi baby had a normal birth weight for 35 weeks' gestation.

Response:

Thank you, we removed this statement and the accompanying reference.

3. It would be helpful to clarify that no sample was available from the mother that would have confirmed mother-to-child transmission. Also, there is no mention of the mother, perhaps because there is no information, but are there any CD4 or viral load data? The mother of the Mississippi baby had a healthy CD4 count (644/mm³) and a low viral load (~2,000 c/ml).

Response:

We recently obtained an ethics approval to include available maternal data in the report. A maternal CD4 count from 2008 was 108 cells/ μ l and from 2010 was 129 cell/ μ l. Regrettably, the mother demised thereafter and without samples for storage. We have now included maternal CD4 data; these are the only data that we have managed to retrieve on the mother.

We have also stated that the mother is deceased and that no stored samples are available from the mother – in the 'methods'.

4. It may be helpful to show this child's initial CD4, CD4% and viral load in the context of the other 226 CHER children who received early ART. This would show where the CD4% in this child is among the group at 8 wks' age: 41.6% sounds very high but readers may not be aware that an entry criterion was a CD4% of >25%. In addition, the comparisons shown with age-matched children and adults, whilst of interest, would be more telling if they could be made with others in the CHER cohort prior to ART interruption. However I imagine the samples are not available for such comparisons.

Response:

Thank you for this suggestion. We have included text describing the child's CD4 count and CD4% in the context of the other early treated CHER children (n=226; n=227 with the child). Most children had very high viral loads reported as >750,000 copies/ml; comparison is not informative so we have not mentioned this in the text.

5. Finally, the sex of the child may have some bearing on the outcome and there is no mention of this. It would be helpful to explain that permission for this information to be included was not provided in this case.

Response:

We have also now obtained assent/consent to include the sex (male) of the child in the report and highlighted that this is the first report of a boy with remission. This is an important fact, as we agree that sex differences can indeed contribute to different outcomes.

Reviewer #3 (Remarks to the Author):

The authors report on a child that was perinatally infected with HIV-1 and received early limited (40 weeks) antiretroviral Treatment as part of the children with early antiRetroviral therapy (CHER) Trial. At Age 9.5 years, diagnostic tests for HIV were negative. Characteristics such as CD4:CD8 Ratio, low T-cell activation and low CCR5 Expression were noted, resembling those of uninfected children. Virus persisted (HIV-1 DNA and Plasma RNA) as measured by sensitive assays but replication-competent Virus was not detected.

This is a very interesting case of a posttreatment Controller. Those patients are rarely identified and have given rise to many studies that want to identify factors explaining posttreatment control.

Although the authors have undertaken many attempts to identify factors explaining this very rare phenomenon, they cannot really do this as in most case reports this is intrinsically difficult with a case of one and due to the heterogeneity of all these cases, large studies will probably never be possible. What is interesting is that the child's virus before treatment replicated a lot and thus was clearly replication competent at that time. Maybe they could also discuss the very recent paper by Colby et al, Nat Med, 2018 Jun 11. doi: 10.1038/s41591-018-0026-6, which did not find any posttreatment control in Fiebig 1 and 2 treated adult patients. Thus, it seems that some replication of the virus is needed for posttreatment control.

Response:

We agree that one cannot extrapolate from one case (or only a few cases) on the factors leading to a state of remission in a few versus the many that cannot achieve remission under similar circumstances. However investigating specific viral or host features of individual cases and testing their importance in other treated or untreated individuals, should increase our understanding of this phenomenon.

We thank the reviewer for pointing out a possible role for limited viral replication. We included in paragraph 2 of the 'discussion' that some viral replication may be important for more durable post-treatment control and have cited Colby et al (2017) which shows with very early treatment (with 10 days of infection in the case studied) and with complete lack of detection of virus (also shown in the paper by Henrich et al, 2017) that viral rebound of virus still occurs .

It also seems to be clear that cellular adaptive immune response is not the major reason for containing the virus. Neutralizing activity of the plasma was not tested as far as I can infer from the manuscript but most likely that was also not the case because neutralizing antibodies rarely seem to be of importance in these post treatment controllers and time for eliciting bNabs before Treatment was too short anyway. An interesting finding is the very low expression of CCR5. Host genetic factors as measured do not explain much.

Taken together, this is an interesting case report, which however, still leaves open, why this is a posttreatment Controller.

What would be interesting to look at further is:

- 1) Full length genome sequencing of the virus (might be difficult to get at these low copy numbers). This would probably be the most interesting additional test. The big question is, whether the Virus is really replication competent or not. It is interesting that Plasma

RNA is still found at very low Levels, thus, Virus particles must be produced (not just transcripts because they would be degraded rapidly in the Plasma). Could there be a possibility that there are Long-lived cells producing viruses with replication defective particles?

- 2) exome sequencing of the host (maybe there is some very rare variant of relevance to be found)
- 3) cellular HIV-RNA (at least unspliced)
- 4) neutralizing antibody Responses if possible

Response:

We appreciate the reviewer's comments and suggestions.

Establishing the likelihood of replication-competence through sequencing full length genomes of the child's provirus is in progress; we are also developing methodology to look at transcription- and translation-competent virus within specific T cell subsets at the single cell level. The multiple approaches to detect replication competence that include detecting various forms of RNA and protein will provide a comprehensive description of events within the child's CD4 T cells. We have already discussed the likelihood of defective particles being produced and have hypothesized that the proviral reservoir may be maintained through homeostatic proliferation: we intend establishing studies to address this.

We agree with the reviewer that there would be insufficient time pre-ART to allow for neutralizing antibody responses to develop. Such responses usually associate with very high viral loads (progressive infection) and/or long duration of viraemia.

We will continue to explore host genetic variation in the child, however we do anticipate that a single variant alone is unlikely to explain the outcome. Complex diseases usually involve combinations of gene variants. Whole human genome sequencing (WGS) is planned if we obtain informed consent (rather than exome sequencing). This will catalogue the variation in this child that will be informative on a gene-per-gene basis as genes of interest become evident in the context of HIV-1 control or remission (children or adult studies). As there are an estimated 20,000 exons (protein-coding regions) in the human genome, interrogating a single human exome or genome (introns included) is insufficient to draw conclusions about a single case of post-treatment control.

Minor comments:

Ref 1, should be completed by Finzi, et al, 1997, Science and Wong et al, Science.

Response:

We have included the 2 references, together with reference 1 – following the sentence of the introduction.

Reviewers' Comments:

Reviewer #1:

Remarks to the Author:

The authors have provided several arguments to address the issues that I raised during the review of the original version of the manuscript describing this very interesting case of posttreatment control in a child. I appreciate the effort that has been made by the authors to provide new information and to modulate their discussion. I still believe that the interpretation of the results obtained (in particular regarding T cell and NK cell response, and viral reactivation assays) is difficult in the absence of any comparison to other children in the CHER study or who initiated antiretroviral therapy in similar conditions to this child. In any case, I agree with the other reviewers and the authors that no formal conclusion about the mechanisms of control can be driven from the study of a single case.

Reviewer #2:

Remarks to the Author:

The authors have made some helpful additions to the manuscript in response to the minor comments made, but have not really addressed the major specific comments. This is such an interesting case, but it would be a lot more valuable to the field if it were to shed light on the underlying mechanism (or indeed on what is not the underlying mechanism), and the opportunities are there to do this in this case.

These included:

1. Sequencing the virus and determining the viral replicative capacity - this would potentially be very informative in providing an indication of the mechanism operating here. The authors agree that this is in their future plans.
2. The finding of a Gag-specific CD4 response at age 9.5yrs in the child could be explored by studying what responses were generated at 58 weeks (after TI), again with a view to exploring mechanism. The authors again agree that this would be part of their future plans.

With respect to the other responses and the revised manuscript:

1. The authors have revised the CD4% from the 58 week timepoint down from 65% to 43%. This (43%) is more likely to be the true figure. Although it has to be said that this might raise the question of the quality of other data presented, I am not myself concerned on this point.
2. The inclusion of adult reference values for NK responses in the Figure 3 legend are helpful if adults and children are similar, but often (as the authors argue in the text) they are different. It would be more relevant, therefore, to show data for age-matched children. Also, are these reference values shown derived from HIV-uninfected individuals?
3. The inclusion of IQRs for the CHER cohort is very helpful. However in the text it is stated that "these values fell into the respective 50th centiles...in the CHER trial". This should be IQR - 50th centile does not mean the same thing as interquartile range.

Reviewer – Philip Goulder

Reviewer #3:

Remarks to the Author:

The authors did pretty much what they could do to address the comments. It would have been nice to add full length sequencing data of the Virus to this case.

Manuscript: NCOMMS-18-14830A

Title: **A child with perinatal HIV infection and long-term sustained virological control following antiretroviral treatment cessation**

Response to reviewers' comments (reviewer 2 and 3 – request for full sequence of virus):

Sequencing of virus – detailed information as to why this is beyond the scope of this manuscript:

The main request following from the reviewer 2 and 3 is for a full-length virus sequence. Firstly, I think it is important to distinguish between virus and integrated provirus DNA within host DNA. The request (clarified by discussions with the senior editor) was centred around the time point of 9.5 years when the *gag* sequencing was conducted. The partial *gag* sequence we provided was from population-based sequencing of nested PCR-amplified HIV DNA (provirus) using primers targeting gag only, and with the purpose of subtyping the virus.

Therefore this sequence is not from free virions but rather of cell-associated HIV DNA (integrated and unintegrated). It is important to note that HIV DNA detected in all HIV-infected individuals is made up mostly of defective proviruses (with large deletions, mutations that impair viral replication) – thought to be >90% of the integrated proviruses (often referred to as the “provirus graveyard”). It is estimated that only 1% might be provirus capable of producing replication-competent virus. A further point to note is that virus sequences found in circulating CD4 cells might not always be the same virus that we are measuring in the peripheral circulation – it is possible that virions are being produced from other tissue/organ sites. This has been shown in some studies looking at rebounding virus after antiretroviral treatment is stopped.

We had in the previous response to reviewers stated the following “We already know that very small amounts of virus are being produced *in vivo* (6.6 RNA copies/ml). In order to obtain sufficient virus for full-length sequencing, we are now storing blood specimens from multiple visits. This will require a number of years but will be a very important comparison well worth the effort”. The rate-limiting step in terms of sensitivity of detection of plasma virus lies at the level of reverse transcribing RNA extracted from the virions; therefore substantial volumes of plasma will be required. This is not doable in any reasonable time. This virus produced *in vivo* is crucial to understanding events in this child.

In our opinion, the detailed description of provirus/virus should be a completely separate paper in order to do the subject justice – this is a large undertaking and will take a substantial amount of time (1 to 2 years). We acquired some funding for this purpose and are developing and applying a subtype C-specific Full-Length Individual Provirus Sequencing (FLIPS) assay, a method that involves diluting out large numbers of CD4 cells (obtained from multiple visits so that we can **amplify from single templates** in order to comprehensively study the proviral DNA landscape

We hypothesise some proviral templates may be able to produce virus proteins but cannot produce whole virus, while some other templates must be packaged into virions to account for the small amount we see produced *in vivo* (6.6 copies/ml of RNA viral load), or alternatively virus may be produced from other tissue/organ sites (e.g. from tissue resident T cells) and bear no resemblance to the integrated provirus in circulating CD4 T cells. We must tease apart all these possibilities to better understand the virus produced currently and to establish if it has been sufficiently “silenced” to account for remission. For example, we may find only “dead-end” versions of virus in circulating T

cells with no ability to replicate in our >9.5 year analysis – if so, this could account for state of remission (assuming sufficiently large numbers of CD4 cells have been tested). Yet how do we explain the virions produced in plasma? We think they might be defective (addressed in the discussion), but these could be replication-competent and be produced from resident T cells or other tissue cells/sites. If so, our interpretation of results of provirus at >9.5 years would be incorrect. Therefore, we need to study the virus reservoir as a whole story – from start to finish.

We plan to address the following questions about the viral reservoir: What does the original virus look like (at early stage when the viral load was indicative of highly replicating virus), and what does it look like now (>9.5 years). Can we get clues as to what happened to the virus between these time points from sequencing the HIV DNA reservoir over time. In addition we will investigate which immune responses might be/have been relevant from footprints in viral sequences.

The importance of the technology and approach employed in deriving full viral genome sequences:

As mentioned, the partial *gag* sequence we provided was from population-based sequencing of nested PCR-amplified HIV DNA (provirus) with the sole purpose of subtyping the virus. It was not our intention to analyse these sequences in any other way at this stage. This sequencing approach is known as single-proviral or single-genome sequencing (SPS/SGS) – where one genetically characterizes a sub-genomic region of the HIV-1 genome. Completing sequencing of the other sub-genomic regions making up the HIV-1 genome to derive a full sequence has major limitations as will not accurately capture the overall diversity and replication competency potential of the HIV-1 proviruses. Reporting a representative provirus sequence that is not accurate cannot inform mechanisms to explain this child’s remission.

The other approach of sequencing (population-based) whole HIV-1 provirus would involve using multiple internal sequencing primers (> 70 primers) that carry the risk of erroneously identifying defective proviruses and make resolving the entire proviral sequence technically challenging. This method is hampered by primer mismatches, and because of the numbers and complexity of primers required, may not capture the entire population of proviruses.

Neither of these 2 approaches provide accuracy of information to answer the question that we all want answered (which would be the point of getting these data) – which is can we find full length proviruses that are capable of producing replication-competent virus.

Both approaches above have limitations, as you are “piecing together” sequences rather than sequencing individual full templates. PCR amplification of mixed templates can also result in recombination events between different templates. **The FLIPS assay we are employing overcomes these limitations** – one generates multiple proviruses from single templates derived from diluting out millions of CD4 cells (so here we are looking for at least 100 individual templates and possibly more depending on the findings, that will individually be amplified and sequenced). So amplifications and sequencing are on quite a scale. We need to also analyse in depth to ensure we have tested sufficient CD4 cells as the reservoir is likely to be very small. Because we are doing this on later samples we can pool samples from different visits. This is improved sequencing technology that is specifically being applied to study the HIV reservoir because of limitations in the methods I outlined above.

Each provirus sequence (>100 individual sequences) will be analysed for the presence of genetic features that render HIV-1 replication defective – these include inversions, large deletions, deleterious stop codons and hypermutations, frameshift mutations, and defects in the packaging

signal or major splice donor (MSD) site. Proportions of proviruses that display any of these defects will be considered defective, while intact proviruses will be considered as potentially replication-competent. Evaluating composition of proviral variants/populations will establish to what extent genetically intact proviruses potentially capable of producing virus populate the reservoir.

Aside from describing genetic features that may associate with defective viral replicative function, we will also determine whether inferred protein sequences across the genome contain HLA class I-specific immune escape mutations from Cytotoxic T cell (CTL) surveillance, and escape from HLA class II (CD4 T cell responses). We will also consider viral escape from other immune responses e.g. HIV-specific antibodies and NK cells.

Reviewers' comments:

Reviewer #1 (Remarks to the Author):

The authors have provided several arguments to address the issues that I raised during the review of the original version of the manuscript describing this very interesting case of posttreatment control in a child. I appreciate the effort that has been made by the authors to provide new information and to modulate their discussion. I still believe that the interpretation of the results obtained (in particular regarding T cell and NK cell response, and viral reactivation assays) is difficult in the absence of any comparison to other children in the CHER study or who initiated antiretroviral therapy in similar conditions to this child. In any case, I agree with the other reviewers and the authors that no formal conclusion about the mechanisms of control can be driven from the study of a single case.

We agree with all of the points made by the reviewer. The mechanisms involved are likely complex and multi-factorial, but with the careful approach we are taking to systematically study the child in detail, we believe we can obtain further insights in time through the study of other cohorts of early-treated HIV-infected children (CHER/other). We agree that no formal conclusion can be drawn about mechanisms of control, however this is true for any studies already published on the rare cases of post-treatment control.

Reviewer #2 (Remarks to the Author):

The authors have made some helpful additions to the manuscript in response to the minor comments made, but have not really addressed the major specific comments. This is such an interesting case, but it would be a lot more valuable to the field if it were to shed light on the underlying mechanism (or indeed on what is not the underlying mechanism), and the opportunities are there to do this in this case.

These included:

1. Sequencing the virus and determining the viral replicative capacity - this would potentially be very informative in providing an indication of the mechanism operating here. The authors agree that this is in their future plans.

Response:

See response under **“Sequencing of virus – detailed information as to why this is beyond the scope of this manuscript”**. Only once all sequencing is completed and we have insights from this, would we only then conduct functional assays to determine replicative capacity.

2. The finding of a Gag-specific CD4 response at age 9.5yrs in the child could be explored by studying what responses were generated at 58 weeks (after TI), again with a view to exploring mechanism. The authors again agree that this would be part of their future plans.

Response:

Our previous response may have been confusing – as the future plans we alluded to were around expanding and studying the CD4 Gag response at 9.5 years. We also stated “Sample availability and volumes at earlier time points make it unlikely we will be able to effectively study all these parameters – as mentioned earlier we would rationalize the use of any such samples very carefully.” We had mentioned this in response to the previous reviewer comments about the rise in CD4% at 58 weeks (values now corrected).

There is no PBMC sample available to do this at the 58 week time point. Regardless, we do not want to commence any study of cellular immune responses at any early time points until we have all the information from virus/provirus sequencing - we will then be able to see what immune pressure the virus has been under (CD4 and CD8). Other responses might also be important, so would want to target answering the question of what immune responses may have been instrumental in the outcome more broadly. We do not anticipate that only one specific type of response is involved in this outcome and think that we should carefully consider our approach, and as suggested by reviewers would be best done in comparison with other early treated children – all extensive studies in their own right.

With respect to the other responses and the revised manuscript:

1. The authors have revised the CD4% from the 58 week timepoint down from 65% to 43%. This (43%) is more likely to be the true figure. Although it has to be said that this might raise the question of the quality of other data presented, I am not myself concerned on this point.

Response:

I think we can all appreciate that one will occasionally encounter errors in analysis or reporting of diagnostic results. What was remarkable was the fact that we could still go back to raw flow cytometry data from 10 years ago, which are all very well catalogued and stored, that allowed us the opportunity to request a re-analysis of the flow data.

2. The inclusion of adult reference values for NK responses in the Figure 3 legend are helpful if adults and children are similar, but often (as the authors argue in the text) they are different. It would be more relevant, therefore, to show data for age-matched children. Also, are these reference values shown derived from HIV-uninfected individuals?

Response:

We are merely using existing reference values from adults (as we have none available for children of different ages) to highlight that the child has adult-like ability (clearly not deficient) as added information. This not a major result but gives a bit more insight into an existing immune cell capability. If the child had very much reduced capability relative to the adult values - this would then beg the question as to whether this is because of age (i.e. children are different to adults and the

child has child-like ability) or because the child has very low NK or T cell responsiveness. This would then warrant trying to establish the respective cell capacities of 10 year old uninfected children. This would not add anything to our findings. However, we could remove Figure 3C but do feel strongly that retaining this extra information is preferable.

We have added to the legend for Figure 3 that the reference values are from HIV-uninfected adults.

3. The inclusion of IQRs for the CHER cohort is very helpful. However in the text it is stated that “these values fell into the respective 50th centiles...in the CHER trial”. This should be IQR - 50th centile does not mean the same thing as interquartile range.

Response:

Thanks for pointing out, this is correct. We have changed ‘These values fell within the respective 50th centiles for all early treated children who stopped ART in the CHER trial – n= 227; median CD4 (IQR): 2,255 (1759-2972); median CD4% 36.4 (31.4-42.5).’ to ‘These values fell within the respective IQRs for all early treated children who stopped ART in the CHER trial – n= 227; median CD4 : 2,255 (IQR: 1759-2972); median CD4% 36.4 (IQR: 31.4-42.5).

Reviewer #3 (Remarks to the Author):

The authors did pretty much what they could do to address the comments. It would have been nice to add full length sequencing data of the Virus to this case.

Response:

See response under “**Sequencing of virus – detailed information as to why this is beyond the scope of this manuscript**”.

We have added the following sentence to the concluding paragraph:” Events that may have led to sufficient “silencing” of virus replication will be explored by whole genome sequencing of virus/provirus.”